# Algorithmic Regularization in Tensor Optimization: Towards a Lifted Approach in Matrix Sensing

**Ziye Ma**
Department of EECS
UC Berkeley
ziyema@berkeley.edu

**Javad Lavaei**
Department of IEOR
UC Berkeley
lavaei@berkeley.edu

**Somayeh Sojoudi**
Department of EECS, ME
UC Berkeley
sojoudi@berkeley.edu

## Abstract

Gradient descent (GD) is crucial for generalization in machine learning models, as it induces implicit regularization, promoting compact representations. In this work, we examine the role of GD in inducing implicit regularization for tensor optimization, particularly within the context of the lifted matrix sensing framework. This framework has been recently proposed to address the non-convex matrix sensing problem by transforming spurious solutions into strict saddles when optimizing over symmetric, rank-1 tensors. We show that, with sufficiently small initialization scale, GD applied to this lifted problem results in approximate rank-1 tensors and critical points with escape directions. Our findings underscore the significance of the tensor parametrization of matrix sensing, in combination with first-order methods, in achieving global optimality in such problems.

## 1 Introduction

This paper is dedicated to addressing the non-convex problem of matrix sensing, which has numerous practical applications and is rich in theoretical implications. Its canonical form can be written as:

$$
\begin{aligned}
&\text{find} \quad M \in \mathbb{R}^{n \times n} \\
&\text{s.t.} \quad \mathcal{A}(M) = \mathcal{A}(M^*) \quad \text{rank}(M) \leq r, M \succeq 0.
\end{aligned}
\tag{1}
$$

$\mathcal{A}(\cdot) : \mathbb{R}^{n \times n} \mapsto \mathbb{R}^m$ is a linear operating consisting of $m$ sensing matrices $\{A_i\}_{i=1}^m \in \mathbb{R}^{n \times n}$ where $\mathcal{A}(M) = [\langle A_1, M \rangle, \dots, \langle A_m, M \rangle]^T$. The sensing matrices and the measurements $b = \mathcal{A}(M^*)$ are given, while $M^*$ is an unknown low-rank matrix to be recovered from the measurements. The true rank of $M^*$ is bounded by $r$, usually much smaller than the problem size $n$. More importantly, since $\mathcal{A}$ is linear, one can replace $A_i$ with $(A_i + A_i^\top)/2$ without changing $b$, and therefore all sensing matrices can be assumed to be symmetric.

The aforementioned problem serves as an extension of both compressed sensing [1], which is widely applied in the field of medical imaging, and matrix completion [2, 3], which possesses an array of notable applications [4]. Additionally, this problem emerges in a variety of real-world situations such as phase retrieval [5–7], motion detection [8], and power system state estimation [9, 10]. A recent study by [11] established that any polynomial optimization problem can be converted into a series of problems following the structure of (1), thereby underscoring the significance of investigating this specific non-convex formulation. Within the realm of contemporary machine learning, (1) holds relevance as it is equivalent to the training problem for a two-layer neural network with quadratic activations [12]. In this context, $m$ denotes the number of training samples, $r$ is the size of the hidden layer, and the sensing matrices $A_i = x_i x_i^\top$ are rank-1, with $x_i$ representing the $i^{th}$ datapoint.

To solve (1), an increasingly popular approach is the Burer-Monteiro (BM) factorization [13], in which the low-rank matrix $M$ is factorized into $M = XX^\top$ with $X \in \mathbb{R}^{n \times r}$, thereby omitting the

37th Conference on Neural Information Processing Systems (NeurIPS 2023).

constraint, making it amenable to simple first-order methods such as gradient descent (GD), while scaling with $\mathcal{O}(nr)$ instead of $\mathcal{O}(n^2)$. The formulation can be formally stated as follows:

$$\min_{X \in \mathbb{R}^{n \times r}} f(X) := \frac{1}{2}\|\mathcal{A}(XX^T) - b\|^2 = \frac{1}{2}\|\mathcal{A}(XX^T - ZZ^\top)\|^2 \quad \text{(Unlifted Problem)} \quad (2)$$

with $Z \in \mathbb{R}^{n \times r}$ being any ground truth representation such that $M^* = ZZ^\top$. Since (2) is a non-convex problem, it can have spurious local minima[1], making it difficult to recover $M^*$ in general. The pivotal concept in solving (1) and (2) to optimality is the notion of Restricted Isometry Property (RIP), which measures the proximity between $\|\mathcal{A}(M)\|_F^2$ and $\|M\|_F^2$ for all low-rank matrices $M$. This proximity is captured by a constant $\delta_p$, where $\delta_p = 0$ means $\mathcal{A}(M) = M$ for matrices up to rank $p$, leading to exact isometry case, and $\delta_p \to 1$ implying a problematic scenario in which the proximity error is large. For a precise definition, please refer to Appendix A.3.

Conventional wisdom suggests that there is a sharp bound on the RIP constant that controls the recoverability of $M^*$, with $1/2$ being the bound for (2). [14, 15] prove that if $\delta_{2r} < 1/2$, then all local minimizers are global minimizers, and conversely if $\delta_{2r} \geq 1/2$, counterexamples can be easily established. Similar bounds of $1/3$ are also derived for general objectives [16, 17], demonstrating the importance of the notion of RIP. However, more recent studies reveal that the technique of over-parametrization (by using $X \in \mathbb{R}^{n \times r_{\text{search}}}$ instead, with $r_{\text{search}} > r$) can take the sharp RIP bound to higher values [18, 19]. Recently, it has also been shown that using a semidefinite programming (SDP) formulation (convex relaxation) can lead to guaranteed recovery with a larger RIP bound that approaches 1 in the transition to the high-rank regime when $n \approx 2r$ [20]. These works all show the efficacy of over-parametrization, shedding light on a powerful way to find the global solution of complex non-convex problems. However, all of these techniques fail to handle real-world cases with $\delta_{2r} \to 1$ in the low-rank regime. To this end, a recent work [21] drew on important concepts from the celebrated Lasserre's Hierarchy [22] and proposed a lifted framework based on tensor optimization that could convert spurious local minimizers of (2) into strict saddle points in the lifted space, for arbitrary RIP constants in the $r = 1$ case. We state this lifted problem below:

$$\min_{\mathbf{w} \in \mathbb{R}^{n \circ l}} \quad \|\langle \mathbf{A}^{\otimes l}, \mathbf{w} \otimes \mathbf{w} \rangle - b^{\otimes l}\|_F^2 \quad \text{(Lifted Problem, } r = 1) \quad (3)$$

where $\mathbf{w}$ is an $l$-way, $n$-dimensional tensor, and $\mathbf{A}^{\otimes l}$ and $b^{\otimes l}$ are tensors "lifted" from $\mathcal{A}$ and $b$ via tensor outer product. We defer the precise definition of tensors and their products to Section 2. The main theorem of [21] states that when $r = 1$, for some appropriate $l$, the first-order points (FOP) of (2) will be converted to FOPs of (3) via lifting, and that spurious second order points (SOP) of (2) will be converted into strict saddles, under some technical conditions, provided that $\mathbf{w}$ is symmetric, and rank-1. This rank-1 constraint on the decision variable $\mathbf{w}$ is non-trivial, since finding the dominant rank-1 component of symmetric tensors is itself a non-convex problem in general, and requires a number of assumptions for it to be provably correct [23, 24]. This does not even account for the difficulties of maintaining the symmetric properties of tensors, which also has no natural guarantees. Therefore, although this lifted formulation may be promising in the pursuit of global minimum, there are still major questions to be answered. Most importantly, it is desirable to know *whether the symmetric, rank-1 condition is necessary, and if so, how to achieve it without explicit constraints?*

The necessity of the condition in question can be better understood through insights from [25]. The authors argue that over-parametrizing non-convex optimization problems can reshape the optimization landscape, with the effect being largely independent of the cost function and primarily determined by the parametrization. This notion is consistent with [21], which contends that over-parametrizing vectors into tensors can transform spurious local solutions into strict saddles. However, [25] specifically examines the parametrization from vectors/matrices to tensors, concluding that stationary points are not generally preserved under tensor parametrization, contradicting [21]. This implies that the symmetric, rank-1 constraints required in (3) are crucial for the conversion of spurious points.

It is essential to devise a method to encourage tensors to be near rank-1, with implicit regularization as a potential solution. There has been a recent surge in examining the implicit regularization effects in first-order optimization methods, such as gradient descent (GD) and stochastic gradient descent (SGD) [26], which has been well-studied in matrix sensing settings [27–29, 12]. This intriguing observation has prompted us to explore the possible presence of similar implicit regularization in tensor spaces. Our findings indicate that when applying GD to the tensor optimization problem

---

[1]A spurious point satisfies first-order and second-order necessary conditions but is not a global minimum.

([3](#)), an implicit bias can be detected with sufficiently small initialization points. This finding does not directly extend from its matrix counterparts due to the intricate structures of tensors, resulting in a scarcity of useful identities and well-defined concepts for even fundamental properties such as eigenvalues. Furthermore, we show that when initialized at a symmetric tensor, the entire GD trajectory remains symmetric, completing the requirements.

In this paper, we demonstrate that over-parametrization alone does not inherently simplify non-convex problems. However, employing a suitable optimization algorithm offers a remarkably straightforward solution, as this specific algorithm implicitly constrains our search to occur within a compact representation of the over-parametrized space without necessitating manual embeddings or transformations. This insight further encourages the investigation of a (parametrization, algorithm) pair for solving non-convex problems, thereby enhancing our understanding of achieving global optimality in non-convex problems.

## 1.1 Related Works

**Over-parametrization in matrix sensing**. Except for the lifting formulation ([3](#)), there are two mainstream approaches to over-parametrization in matrix sensing. The first one is done via searching over $Y \in \mathbb{R}^{n \times r_{\text{search}}}$ instead of $X \in \mathbb{R}^{n \times r}$, and using some distance metric to minimize the distance between $\mathcal{A}(YY^\top)$ and $b$. Using an $l_2$ norm, [18, 19] established that if $r_{\text{search}} > r[(1 + \delta_n)/(1 - \delta_n) - 1]^2/4$, with $r \leq r_{\text{search}} < n$, then every second-order point $\hat{Y} \in \mathbb{R}^{n \times r_{\text{search}}}$ satisfies that $\hat{Y}\hat{Y}^\top = M^*$. [30] showed that even in the over-parametrized regime, noise can only finitely influence the optimization landscape. [29] offered similar results for an $l_1$ loss under good enough RIP constant. Another popular approach to over-paramerization is to use a convex SDP formulation, which is a convex relaxation of ([1](#)) [31]. It has been known for years that as long as $\delta_{2r} < 1/2$, then the global optimality of the SDP formulation correspond to the ground truth $M^*$ [32]. Recently [20] updated this bound to $2r/(n + (n - 2r)(2l - 5))$, which can approach 1 if $n \approx 2r$.

**Algorithm regularization in over-parametrized matrix sensing**. [12, 33] prove that the convergence to global solution via GD is agnostic of $r_{\text{search}}$, in that it only depends on initialization scale, step-size, and RIP property. [29] demonstrates the same effect for an $l_1$ norm, and further showed that a small initialization nullifies the effect of over-parametrization. Besides these works, [27] refined this analysis, showing that via a sufficiently small initialization, the GD trajectory will make the solution implicitly penalize towards rank-$r$ matrices after a small number of steps. [28] took it even further by showing that the GD trajectory will first make the matrix rank-1, rank-2, all the way to rank-$r$, in a sequential way, thereby resembling incremental learning.

**Implicit bias in tensor learning**. The line of work [34–36] demonstrates that for a class of tensor factorization problems, as long as the initialization scale is small, the learned tensor via GD will be approximately rank-1 after an appropriate number of steps. Our paper differs from this line of work in three meaningful ways: 1) The problem considered in those works are optimization problems over vectors, not tensors, and therefore the goal is to learn the structure of a known tensor, rather than learning a tensor itself; 2) Our proof relies directly on tensor algebra instead of adopting a dynamical systems perspective, providing deeper insights into tensor training dynamics while dispensing with the impractical assumption of an infinitesimal step-size.

## 1.2 Main Contributions

1. We demonstrate that, beyond vector and matrix learning problems, optimization of differentiable objectives, such as the $l_2$ norm, through Gradient Descent (GD) can encourage a more compact representation for tensors as decision variables. This results in tensors being approximately rank-1 after a number of gradient steps. To achieve this, we employ an innovative proof technique grounded in tensor algebra and introduce a novel tensor eigenvalue concept, the variational eigenvalue (v-eigenvalue), which may hold independent significance due to its ease of use in optimization contexts.

2. We show that if a tensor is a first-order point of the lifted objective ([3](#)) and is approximately rank-1, then its rank-1 component can be mapped to an FOP of ([2](#)), implying that all FOPs of ([3](#)) lie in a small sphere around the lifted FOPs of ([2](#)). Furthermore, these FOPs possess an escape direction when reasonably distant from the ground truth solution, irrespective of the Restricted Isometry Property (RIP) constants.

3. We present a novel lifted framework that optimizes over symmetric tensors to accommodate the over-parametrization of matrix sensing problems with arbitrary $r$. This approach is necessary because directly extending the work of [21] from $r = 1$ to higher values may lead to non-cubical and, consequently, non-symmetric tensors.

## 2 Preliminaries

Please refer to Appendix A.1 and A.2 for the notations and definitions of first-order and second-order conditions. Here, we introduce two concepts that are critical in understanding our main results.

**Definition 1** (Tensors and Products). We define an $l$-way tensor as:

$$\mathbf{a} = \{a_{i_1 i_2 \dots i_l} | 1 \le i_k \le n_k, 1 \le k \le l\} \in \mathbb{R}^{n_1 \times \dots \times n_l}$$

Moreover, if $n_1 = \dots = n_l$, then we call this tensor an $l$-order (or $l$-way), $n$-dimensional tensor. $\mathbb{R}^{n \circ l}$ is an abbreviated notion for $n \circ l := n \times \dots \times n$. In this work, tensors are denoted with bold letters unless specified otherwise. The tensor outer product, denoted as $\otimes$, of 2 tensors $\mathbf{a}$ and $\mathbf{b}$, respectively of orders $l$ and $p$, is a tensor of order $l + p$, namely $\mathbf{c} = \mathbf{a} \otimes \mathbf{b}$ with $c_{i_1 \dots i_l j_1 \dots j_p} = a_{i_1 \dots i_l} b_{j_1 \dots j_p}$. We also use the shorthand $\mathbf{a}^{\otimes l}$ for repeated outer product of $l$ times for arbitrary tensor/matrix/vector $\mathbf{a}$. $\langle \mathbf{a}, \mathbf{b} \rangle_{i_1, \dots, i_d}$ denotes tensor inner product along dimensions $i_1, \dots, i_d$ (with respect to the first tensor), in which we simply sum over the specified dimensions after the outer product $\mathbf{a} \otimes \mathbf{b}$ is calculated. This means that the inner product is of $l + p - 2d$ orders. Please refer to Appendix A.3 for a more in-depth review on tensors, especially on its symmetry and rank.

**Definition 2** (Restricted Strong Smoothness (RSS) and Restricted Strong Convexity (RSC)). The linear operator $\mathcal{A} : \mathbb{R}^{n \times n} \mapsto \mathbb{R}^m$ satisfies the $(L_s, r)$-RSS property and the $(\alpha_s, r)$-RSC property if

$$f(M) - f(N) \le \langle M - N, \nabla f(N) \rangle + \frac{L_s}{2} \|M - N\|_F^2$$

$$f(M) - f(N) \ge \langle M - N, \nabla f(N) \rangle + \frac{\alpha_s}{2} \|M - N\|_F^2$$

are satisfied, respectively for all $M, N \in \mathbb{R}^n$ with $\mathrm{rank}(M), \mathrm{rank}(N) \le r$. Note that RSS and RSC provide a more expressible way to represent the RIP property, with $\delta_r = (L_s - \alpha_s)/(L_s + \alpha_s)$.

## 3 The Lifted Formulation for General $r$

A natural extension of (3) to general $r$ requires that instead of optimizing over $X \in \mathbb{R}^{n \times r}$, we optimize over $\mathbb{R}^{[n \times r] \circ l}$ tensors, and simply making tensor outer products between $\mathbf{w}$ to be inner products. However, such a tensor space is non-cubical, and subsequently not symmetric. This is the higher-dimensional analogy of non-square matrices, which lacks a number of desirable properties, as per the matrix scenario. In particular, it is necessary for our approach to optimize over a cubical, symmetric tensor space since in the next section we prove that there exists an implicit bias of the gradient descent algorithm under that setting.

In order to do so, we simply vectorize $X \in \mathbb{R}^{n \times r}$ into $\mathrm{vec}(X) \in \mathbb{R}^{nr}$, and optimize over the tensor space of $\mathbb{R}^{nr \circ l}$, which again is a cubical space. In order to convert a tensor $\mathbf{w} \in \mathbb{R}^{nr \circ l}$ back to $\mathbb{R}^{[n \times r] \circ l}$ to use a meaningful objective, we introduce a new 3-way permutation tensor $\mathbf{P} \in \mathbb{R}^{n \times r \times nr}$ that "unstacks" vectorized matrices. Specifically,

$$\langle \mathbf{P}, \mathrm{vec}(X) \rangle_3 = X \quad \forall X \in \mathbb{R}^{n \times r}, n, r \in \mathbb{Z}^+$$

Such $\mathbf{P}$ can be easily constructed via filling appropriate scalar "1"s in the tensor. Via Lemma A.4, we also know that

$$\langle \mathbf{P}^{\otimes l}, \mathrm{vec}(X)^{\otimes l} \rangle_{3*[l]} = (\langle \mathbf{P}, \mathrm{vec}(X) \rangle_3)^{\otimes l} = X^{\otimes l} \tag{4}$$

where $[l]$ denotes the integer set $[1, \dots, l]$, and $c * [l]$ denotes $[c, 2c, \dots, c * l]$ for some $c \in \mathbb{Z}^+$. For notational convenience, we abbreviate $\langle \mathbf{P}^{\otimes l}, \mathbf{w} \rangle_{3*[l]}$ as $\mathbf{P}(\mathbf{w})$ for any arbitrary $z$-dimensional tensor $\mathbf{w}$ where $z$ can be broken down into the product of two positive integers. Thus, using (4), we can extend (3) to a problem of general $r$, yet still defined over a cubical tensor space:

$$\min_{\mathbf{w} \in \mathbb{R}^{nr \circ l}} \|\langle \mathbf{A}^{\otimes l}, \langle \mathbf{P}(\mathbf{w}), \mathbf{P}(\mathbf{w}) \rangle_{2*[l]} \rangle - b^{\otimes l}\|_F^2 \quad \text{(Lifted formulation, general } r) \tag{5}$$

Let us define a 3-way tensor $\mathbf{A} \in \mathbb{R}^{m \times n \times n}$ so that $\mathbf{A}_{kij} = (A_k)_{ij} \; \forall k \in [m], (i,j) \in [n] \times [n]$. Define $f^l(\cdot) : \mathbb{R}^{n \circ 2l} \mapsto \mathbb{R}$ and $h^l(\cdot) : \mathbb{R}^{[n \times r] \circ l} \mapsto \mathbb{R}$ as $f^l(\mathbf{M}) := \|\langle \mathbf{A}^{\otimes l}, \mathbf{M} \rangle - b^{\otimes l}\|_F^2$ and $h^l(\mathbf{w}) = f^l(\langle \mathbf{w}, \mathbf{w} \rangle_{2*[l]})$, with $\nabla f^l(\cdot) = \nabla_{\mathbf{M}} f^l(\cdot)$ and $\nabla h^l(\cdot) = \nabla_{\mathbf{w}} h^l(\cdot)$.

We prove that (5) has all the good properties detailed in [21] for (3). In particular, we prove that the symmetric, rank-1 FOPs of (5) have a one-to-one correspondence with those of (2), and that those FOPs that are reasonably separated from $M^*$ or have a small $r^{th}$ singular value can be converted to strict saddle points via some level of lifting. For the detailed theorems and proofs, please refer to Appendix B.

# 4 Implicit Bias of Gradient Descent in Tensor Space

In this section, we study why and how applying gradient descent to (5) will result in an implicit bias towards to rank-1 tensors. Prior to presenting the proofs, we shall elucidate the primary intuition behind how GD contributes to the implicit regularization of (2). This will aid in comprehending the impact of implicit bias on (5), as they share several crucial observations, albeit encountering greater technical hurdles. Consider the first gradient step of (2), initialized at a random point $X_0 \in \mathbb{R}^{n \times r_{\text{search}}} = \epsilon X$ with $\|X\|_F^2 = 1$ and $r_{\text{search}} \geq r$:

$$X_1 = X_0 - \eta \nabla h(X_0) = (I + \eta \left[ \mathcal{A}^*\mathcal{A}(M^*) \right]) X_0 - \left[ \mathcal{A}^*\mathcal{A}(X_0 X_0^\top) \right] X_0$$
$$= (I + \eta \left[ \mathcal{A}^*\mathcal{A}(M^*) \right]) X_0 - \epsilon^2 \left[ \mathcal{A}^*\mathcal{A}(X X^\top) \right] X_0$$
$$= (I + \eta \left[ \mathcal{A}^*\mathcal{A}(M^*) \right]) X_0 + \mathcal{O}(\epsilon^3)$$

where $\eta$ is the step-size. Therefore, if $\epsilon$ is chosen to be small enough, we have that

$$X_t \approx (I + \eta \mathcal{A}^*\mathcal{A}(M^*))^t X_0 \quad \text{as } \epsilon \to 0$$

Again, according to the symmetric assumptions on $\mathcal{A}$, we can apply spectral theorem on $\mathcal{A}^*\mathcal{A}(M^*) = \sum_{i=1}^{n} \lambda_i v_i v_i^\top$ for which the eigenvectors are orthogonal to each other. It follows that $X_t \approx \left( \sum_{i=1}^{n} (1 + \eta \lambda_i)^t v_i v_i^\top \right) X_0$.

In many papers surveyed above on making an argument of implicit bias, it is assumed that there is very strong geometric uniformity, or under the context of this paper, it means that $L_s/\alpha_s \approx 1$. Under this assumption, we have $f(M) \approx f(N) + \langle M - N, \nabla f(M) \rangle + \|M - N\|_F^2/2$, leading to the fact that $\nabla^2 f(M) = \mathcal{A}^*\mathcal{A} \approx I$. This immediately gives us $\mathcal{A}^*\mathcal{A}(M^*) \approx M^*$ so that $\lambda_{r+1}, \ldots, \lambda_n \approx 0$ as $M^*$ is by assumption a rank-$r$ matrix. This further implies that $X_t \approx \left( \sum_{i=1}^{r} (1 + \eta \lambda_i)^t v_i v_i^\top \right) X_0$, which will become a rank-r matrix, achieving the effect of implicit regularization, as $X$ is now over-parametrized by having $r_{\text{search}} \geq r$.

However, when tackling the implicit regularization problem in tensor space, one key deviation from the aforementioned procedure is that $L_s/\alpha_s$ will be relatively large, as otherwise there will be no spurious solutions, even in the noisy case [14, 30], which is also the motivation for using a lifted framework in the first place. Therefore, instead of saying that $\mathcal{A}^*\mathcal{A}(M^*) \approx M^*$, we aim to show that the gap between the eigenvalues of a comparable tensor term will enlarge as we increase $l$, making the tensor predominantly rank-1. This observation demonstrates the power of the lifting technique, while at the same time eliminates the critical dependence on a small $L_s/\alpha_s$ factor that is in practice often unachievable due to requiring sample numbers $m$ in the asymptotic regime [37].

Therefore, in order to establish an implicit regularization result for (5), there are four major steps that need to be taken:

1. Proving that a point on the GD trajectory $\mathbf{w}_t$ admits a certain breakdown in the form $\mathbf{w}_t = \langle \mathbf{Z}_t, \mathbf{w}_0 \rangle - \mathbf{E}_t$ for some $\mathbf{Z}_t$ and $\mathbf{E}_t$.

2. Proving that the spectral norm (equivalence of largest singular value) of $\mathbf{E}_t$ is small (scales with initialization scale $\epsilon$)

3. Proving that $\langle \mathbf{Z}_t, \mathbf{w}_0 \rangle$ has a large separation between its largest and second largest eigenvalues using a tensor version of Weyl's inequality.

4. Showing that, with the above holding true, $\mathbf{w}_t$ is predominantly rank-1 after some step $t_*$.

Lemmas 12, 13, 2, and Theorem 1 correspond to the above four steps, respectively. The reader is referred to the lemmas and theorem for more details.

## 4.1 A Primer on Tensor Algebra and Maintaining Symmetric Property

We start with the spectral norm of tensors, which resembles the operator norm of matrices [38].

**Definition 3.** Given a cubic tensor $\mathbf{w} \in \mathbb{R}^{n \circ l}$, its spectral norm $\| \cdot \|_S$ is defined respectively as:

$$\|\mathbf{w}\|_S = \sup \left\{ |\langle \mathbf{w}, u^{\otimes l} \rangle| \; \|u\|_2 = 1, u \in \mathbb{R}^n \right\}$$

There are many definitions for tensor eigenvalues [39], and in this paper we introduce a novel variational characterization of eigenvalues that resembles the Courant-Fisher minimax definition for eigenvalues of matrices, called the v-Eigenvalue. We denote the $i^{th}$ v-Eigenvalue of $\mathbf{w}$ as $\lambda_i^v(\mathbf{w})$. Note this is a new definition that is first introduced in this paper and might be of independent interest outside of the current scope.

**Definition 4** (Variational Eigenvalue of Tensors). For a given tensor $\mathbf{w} \in \mathbb{R}^{n \circ l}$, we define its $k^{th}$ variational eigenvalue (v-Eigenvalue) $\lambda_k^v(\mathbf{w})$ as

$$\lambda_k^v(\mathbf{w}) := \max_{\substack{S \\ \dim(S) = k}} \min_{\mathbf{u} \in S} \frac{|\langle \mathbf{w}, \mathbf{u} \rangle|}{\|\mathbf{u}\|_F^2}, \quad k \in [n]$$

where $S$ is a subspace of $\mathbb{R}^{n \circ l}$ that is spanned by a set of orthogonal, symmetric, rank-1 tensors. Its dimension denotes the number of orthogonal tensors that span this space. It is apparent from the definition that $\|\mathbf{w}\|_S = \lambda_1^v(\mathbf{w})$.

Next, since most of our analysis relies on the symmetry of the underlying tensor, it is desirable to show that every tensor along the optimization trajectory of GD on (5) remains symmetric if started from a symmetric tensor. Please find its proof in Appendix C.2.

**Lemma 1.** If the GD trajectory of (5) $\{\mathbf{w}_t\}_{t=0}^{\infty}$ is initialized at a symmetric rank-1 tensor $\mathbf{w}_0$, then $\{\mathbf{w}_t\}_{t=0}^{\infty}$ will all be symmetric.

## 4.2 Main Ideas and Proof Sketch

In this subsection, we highlight the main ideas behind implicit bias in GD. Lemma 12 and 13 details the first and second step, and are deferred to Appendix C.2. The proofs to the results of this section can also be found in that appendix. The lemmas alongside with their proofs are highly technical and not particularly enlightening, therefore omitted here for simplicity. However, the most important takeaway is that for the $t^{th}$ iterate along the GD trajectory of (5), we have the decomposition

$$\mathbf{w}_{t+1} = \langle \mathbf{Z}_t, \mathbf{w}_0 \rangle - \mathbf{E}_t := \tilde{\mathbf{w}}_t - \mathbf{E}_t$$

for some $\mathbf{Z}_t$ and $\mathbf{E}_t$ such that $\|\mathbf{E}_t\|_S = \mathcal{O}(\epsilon^3)$. This essentially means that by scaling the initialization $\mathbf{w}_0$ to be small in scale, the error term $\mathbf{E}_t$ can be ignored from a spectral standpoint, and scales with $\epsilon$ at a cubic rate. This will soon be proven to be useful next.

**Lemma 2.** Given $\mathbf{w}_t$ along the GD trajectory of (5), its first two v-eigenvalues, as defined in definition 4, satisfy the relation

$$\frac{\lambda_2^v(\mathbf{w}_t)}{\lambda_1^v(\mathbf{w}_t)} \leq \frac{\|x_0\|_2^l (1 + \eta \sigma_2^l(U))^t + \|\mathbf{E}_t\|_S/\epsilon}{|v_1^\top x_0|^l (1 + \eta \sigma_1^l(U))^t - \|\mathbf{E}_t\|_S/\epsilon} = \frac{\|x_0\|_2^l (1 + \eta \sigma_2^l(U))^t + \mathcal{O}(\epsilon^2)}{|v_1^\top x_0|^l (1 + \eta \sigma_1^l(U))^t - \mathcal{O}(\epsilon^2)} \quad (6)$$

where $\sigma_1(U)$ and $\sigma_2(U)$ denote the first and second singular values of $U = \langle \mathbf{A}_r^* \mathbf{A}, M^* \rangle \in \mathbb{R}^{nr \times nr}$, and $v_1, v_2$ are the associated singular vectors.

Lemma 2 showcases that when $\epsilon$ is small, the ratio between the largest and second largest v-eigenvalues of $\mathbf{w}$ is dominated by $(\|x_0\|_2^l (1 + \eta \sigma_2^l(U))^t)/(|v_1^\top x_0|^l (1 + \eta \sigma_1^l(U))^t)$.

Now, if either $\|x_0\|_2^l$ is large or $|v_1^\top x_0|^l$ approaches 0 in value, then the ratio may be relatively large, contradicting our claim. However, this issue can be easily addressed by letting $x_0 = v_1 + g \in \mathbb{R}^{nr}$, where $g$ is a vector with each entry being i.i.d sampled from the Gaussian distribution $\mathcal{N}(0, \rho)$. Note that since $U = \langle \mathbf{A}_r, b \rangle_3$, we can calculate $U$ and $v_1$ directly. Lemma 14 in Appendix C.2 shows that with this initialization, $|v_1^\top x_0|^l = \mathcal{O}(1)$ and $\|x_0\|_2^l = \mathcal{O}(1)$ with high probability if we select $\rho = \mathcal{O}(1/nr)$. Therefore, the $t^{th}$ iterate along the GD trajectory of (5) satisfies

$$\frac{\lambda_2^v(\mathbf{w}_t)}{\lambda_1^v(\mathbf{w}_t)} \lesssim \frac{(1 + \eta \sigma_2^l(U))^t}{(1 + \eta \sigma_1^l(U))^t} \quad (7)$$

with hight probability if $\rho$ is small. This implies that "the level of parametrization helps with separation of eigenvalues", since increasing $l$ will decrease ratio $\lambda_2^v(\mathbf{w}_t)/\lambda_1^v(\mathbf{w}_t)$. Furthermore, regardless of the value of $\sigma_1(U)$, a larger $t$ will make this ratio exponentially smaller, proving the efficacy of algorithmic regularization of GD in tensor space.

By combining the above facts, we arrive at a major result showing how a small initialization could make the points along the GD trajectory penalize towards rank-1 as $t$ increases

**Theorem 1.** *Given the optimization problem* (5) *and its GD trajectory over some finite horizon* $T$, *i.e.,* $\{\mathbf{w}_t\}_{t=0}^T$ *with* $\mathbf{w}_{t+1} = \mathbf{w}_t - \eta \nabla h^l(\mathbf{w}_t)$, *where* $\eta$ *is the stepsize, then there exist* $t(\kappa, l) \geq 1$ *and* $\kappa < 1$ *such that*

$$\frac{\lambda_2^v(\mathbf{w}_t)}{\lambda_1^v(\mathbf{w}_t)} \leq \kappa, \qquad \forall t \in [t(\kappa, l), t_T] \tag{8}$$

*if* $\mathbf{w}_0$ *is initialized as* $\mathbf{w}_0 = \epsilon x_0^{\otimes l}$ *with a sufficiently small* $\epsilon$, *where* $t(\kappa, l)$ *is expressed as*

$$t(\kappa, l) = \left\lceil \ln\left(\frac{\|x_0\|_2^l}{\kappa |v_1^\top x_0|^l}\right) \ln\left(\frac{1 + \eta\sigma_1^l(U)}{1 + \eta\sigma_2^l(U)}\right)^{-1} \right\rceil \tag{9}$$

By using the initialization introduced in Lemma 14, we can improve the result of Theoerem 1, which does not need $\epsilon$ to be arbitrarily small. The full details are presented in Corollary 1 in Appendix C.2, stating that as along as $t \asymp \ln(1/\kappa) \ln\left((1 + \eta\sigma_1^l(U))/(1 + \eta\sigma_1^l(U))\right)^{-1}$, $\mathbf{w}_t$ will be $\kappa$-rank-1, as long as $\epsilon$ is chosen as a function of $U, r, n, L_s$, and $\kappa$. Note that we say a tensor $\mathbf{w}$ is "$\kappa$-rank-1" if $\lambda_2^v(\mathbf{w})/\lambda_1^v(\mathbf{w}) \leq \kappa$.

## 5 Approximate Rank-1 Tensors are Benign

Now that we have established the fact that performing gradient descent on (5) will penalize the tensor towards rank-1, it begs the question whether approximate rank-1 tensors can also escape from saddle points, which is the most important question under study in this paper. Please find the proofs to the results in this section in Appendix D.

To do so, we first introduce a *major spectral* decomposition of symmetric tensors that is helpful.

**Proposition 1.** *Given a symmetric tensor* $\mathbf{w} \in \mathbb{R}^{nr \circ l}$, *it can be decomposed into two terms, namely a term consisting of its dominant component and another term that is orthogonal to this direction:*

$$\mathbf{w} = \pm\lambda_1^v(\mathbf{w})w_s^{\otimes l} + \mathbf{w}^\dagger := \mathbf{w}_\sigma + \mathbf{w}^\dagger, \quad w_s \in \mathbb{R}^n, \|w_s\|_2 = 1 \tag{10}$$

*where* $\langle \mathbf{w}, w_s^{\otimes l} \rangle = \lambda_1^v(\mathbf{w})$ *and* $\langle \mathbf{w}^\dagger, w_s^{\otimes l} \rangle = 0$. *Furthermore, if* $\mathbf{w}$ *is a* $\kappa$-rank-1 *tensor, then* $\|\mathbf{w}^\dagger\|_S \leq \kappa\lambda_1^v(\mathbf{w}_t)$.

Next, we characterize the first-order points of (5) with approximate rank-1 tensors in mind. Previously, we showed that if a given FOP of (5) is symmetric and rank-1, it has a one-to-one correspondence with FOPs of (2). However, if the FOPs of (5) are not exactly rank-1, but instead $\kappa$-rank-1, it is essential to understand whether they maintain the previous properties. This will be addressed below.

**Proposition 2.** *Assume that a symmetric tensor* $\mathbf{w} \in \mathbb{R}^{nr \circ l}$ *is an FOP of* (5), *meaning that* (17a) *holds. If it is a* $\kappa$-rank-1 *tensor with* $\kappa \leq \mathcal{O}(1/\|M^*\|_F^2)$, *then it admits a decomposition as*

$$\mathbf{w} = \pm\lambda_1^v(\mathbf{w})\hat{w}^{\otimes l} + \mathbf{w}^\dagger$$

*with* $\mathrm{mat}(\hat{w}) \in \mathbb{R}^{n \times r}$ *being an FOP of* (2) *and* $\|\mathbf{w}^\dagger\|_S \leq \kappa\lambda_1^v(\mathbf{w})$ *by definition.*

The proposition above asserts that for any given FOP of (5), if it is $\kappa$-rank-1 rather than being truly rank-1, it will consist of a rank-1 term representing a lifted version of an unlifted FOP, as well as a term with a small spectral norm. Referring to (58), it is possible to achieve a significantly low $\kappa$ through a moderate number of iterations. This result, considered the cornerstone of this paper, demonstrates that the use of gradient descent with small initialization will find critical points that are lifted FOPs of (2) with added noise, maintaining a robust association between FOPs of (5) and (2). This finding also facilitates this subsequent theorem:

**Theorem 2.** *Assume that a symmetric tensor $\hat{\mathbf{w}} \in \mathbb{R}^{n^r \circ l}$ is an FOP of (5) that is $\kappa$-rank-1 with $\kappa \leq \mathcal{O}(1/\|M^*\|_F^2)$. Consider its major spectral decomposition $\hat{\mathbf{w}} = \lambda_S \hat{x}^{\otimes l} + \hat{\mathbf{w}}^\dagger$ with $\hat{x} \in \mathbb{R}^{nr}$, then it has a rank-1 escape direction if $\hat{X} = \mathrm{mat}(\hat{x})$ satisfies the inequality*

$$\|M^* - \hat{X}\hat{X}^\top\|_F^2 \geq \frac{L_s}{\alpha_s} \lambda_r(\hat{X}\hat{X}^\top)\,\mathrm{tr}(M^*) + \mathcal{O}(r\kappa^{1/l}) \tag{11}$$

*where $l$ is odd and large enough so that $l > 1/(1 - \log_2(2\beta))$ and $\beta$ is defined as*

$$\beta = \frac{L_s\,\mathrm{tr}(M^*)\lambda_r(\hat{X}\hat{X}^\top)}{\alpha_s\|M^* - \hat{X}\hat{X}^\top\|_F^2 - \mathcal{O}(r\kappa^{1/l})}.$$

This theorem conveys the message that by running GD on (5), all critical points have escape directions as long as the point is not close to the ground truth solution. In Appendix B, we present Theorem 7 to provide sufficient conditions for the conversion to hold globally when (11) is hard to hold.

## 6 Numerical Experiments

In this section[2], after we run a given algorithm on (5) to completion and obtain a final tensor $\mathbf{w}_T$, we then apply tensor PCA (detailed in Appendix F) on $\mathbf{w}_T$ to extract its dominant rank-1 component and recover $X_T \in \mathbb{R}^{n \times r}$ such that $(\mathbf{w}_T)_s = \lambda_s\,\mathrm{vec}(X_T)^{\otimes l}$. Since $\mathbf{w}_T$ will be approximately rank-1, the success of this operation is expected [23, 24]. We consider a trial to be successful if the recovered $X_T$ satisfies $\|X_T X_T^\top - M^*\|_F \leq 0.05$. We also initialize our algorithm as per Lemma 14.

### 6.1 Perturbed Matrix Completion

The perturbed matrix completion problem is introduced in [20], which is a noisy version of classic matrix completion problems. The $\mathcal{A}$ operator is introduced as

$$\mathcal{A}_\rho(\mathbf{M})_{ij} := \begin{cases} \mathbf{M}_{ij}, & \text{if } (i,j) \in \Omega \\ \rho\mathbf{M}_{ij}, & \text{otherwise} \end{cases}, \tag{12}$$

where $\Omega$ is a measurement set such that $\Omega = \{(i,i),(i,2k),(2k,i)|\ \forall i \in [n], k \in [\lfloor n/2 \rfloor]\}$. [20] has proved that each such instance has $\mathcal{O}(2^{\lceil n/2 \rceil} - 2)$ spurious local minima, while it satisfies the RIP property with $\delta_{2r} = (1 - \rho)/(1 + \rho)$ for some sufficiently small $\rho$. This implies that common first-order methods fail with high probability for this class of problems. In our experiment, we apply both lifted and unlifted formulations to (12) with $\rho = 0.01$, yielding $\delta_{2r} \approx 1$. We test different values of $n$ and $\epsilon$, using a lifted level of $l = 3$. We ran 10 trials each to calculate success rate. If unspecified in the plot, we default $n = 10$, $\epsilon = 10^{-7}$. Figure 1 reveals a higher success rate for the lifted formulation across different problem sizes, with smaller problems performing better as expected (since larger problems require a higher lifting level). Success rates improve with smaller $\epsilon$, emphasizing the importance of small initialization. We employed customGD, a modified gradient descent algorithm with heuristic saddle escaping. This algorithm will deterministically escape from critical points utilizing knowledge from the proof of Theorem 4. For details please refer to Appendix F. Furthermore, to showcase the implicit penalization affects of GD, we obtained approximate measures for $\lambda_2^v(\mathbf{w}_t)/\lambda_1^v(\mathbf{w}_t)$ (since exactly solving for them is NP-hard) along the trajectory, and presented the results and methods in Appendix E.

Additionally, we examine different algorithms for (5), including customGD, vanilla GD, perturbed GD ([40], for its ability to escape saddles), and ADAM [41]. Figure 2 suggest that ADAM is an effective optimizer with a high success rate and rapid convergence, indicating that momentum acceleration may not hinder implicit regularization and warrants further research. Perturbed GD performed poorly, possibly due to random noise disrupting rank-1 penalization.

### 6.2 Shallow Neural Network Training with Quadratic Activation

It has long been known that the matrix sensing problem (2) includes the training of two-layer neural networks (NN) with quadratic activation as a special case [12]. In summary, the output of the neural

---

[2]'https://github.com/anonpapersbm/implicit_bias_tensor',run on 2021 Macbook Pro

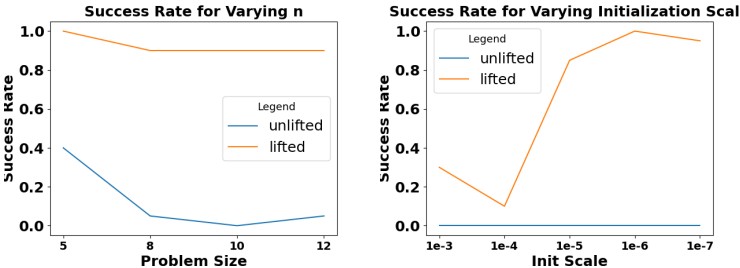

Figure 1: Success rate of the lifted formulation versus the unlifted formulation against varying $n$ and $\epsilon$. The algorithm of choice is CustomGD (details in Appendix F).

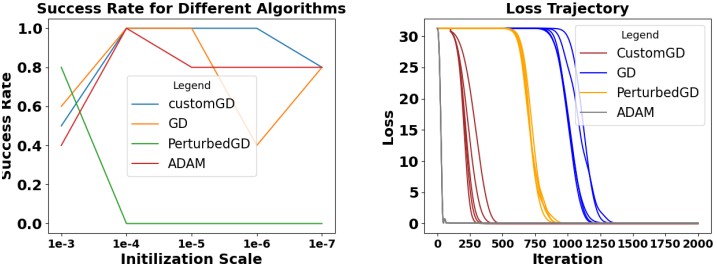

Figure 2: Performance of different algorithms applied to the lifted formulation (5).

network $y \in \mathbb{R}^m$ with respect to $m$ inputs $\{d_i\}_{i=1}^m \in \mathbb{R}^n$ can be expressed as $y_i = \mathbf{1}^\top q(X^\top d_i)$, which implies $y_i = \langle d_i d_i^\top, XX^\top \rangle$, where $q(\cdot)$ is the element-wise quadratic function and $X \in \mathbb{R}^{n \times r}$ in (2) represents the weights of the neural network. Thus $r$ represents the number of hidden neurons. In our experiment, we demonstrate that when $m$ is small, the lifted framework (5) outperforms standard neural network training in success rate, yielding improved recovery of the true weights. We set the hidden neurons number to be $n$ for the standard network training, thereby comparing the existing over-parametrization framework (recall Section 1.1, with $r_{\text{search}} = n$) with the lifted one . We employ the ADAM optimizer for both methods. Table 2 showcases the success rate under various problem and sample sizes. Sampling both data and true weights $Z \in \mathbb{R}^{n \times r}$ from an i.i.d Gaussian distribution, we calculate the observations $y$ and attempt to recover $Z$ using both approaches. As the number of samples increases, so does the success rate, with the lifted approach offering significantly better accuracy overall, even when the standard training has a 0% success rate.

| Success Rate | m = 20 | m=30 | m=40 |
|---|---|---|---|
| n=8 | 0.9(0) | 1(0.3) | 0.9(0.5) |
| n=10 | 0.2(0) | 0.6(0) | 0.8(0) |
| n=12 | 0.1(0) | 0.4(0) | 0.8(0) |

(a) Ground truth weight with $r = 1$

| Success Rate | m = 30 | m=40 | m=50 |
|---|---|---|---|
| n=8 | 0.3(0) | 0.3(0) | 0.8(0) |
| n=10 | 0.3(0) | 0.4(0) | 0.2(0) |
| n=12 | 0(0) | 0(0) | 0.2(0) |

(b) Ground truth weight with $r = 2$

Table 2: Success rate of NN training using (5) and original formulation. The number inside the parentheses denotes the success rate of the original formulations. $\epsilon = 10^{-5}$ and $l = 3$.

# 7 Conclusion

Our study highlights the pivotal role of gradient descent in inducing implicit regularization within tensor optimization, specifically in the context of the lifted matrix sensing framework. We reveal that GD can lead to approximate rank-1 tensors and critical points with escape directions when initialized at an adequately small scale. This work also contributes to the usage of tensors in machine learning models, as we introduce novel concepts and techniques to cope with the intrinsic complexities of tensors.

## 8 Acknowledgement

This work was supported by grants from ARO, ONR, AFOSR, NSF, and the UC Noyce Initiative.

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
