# A Additional Definitions and Supporting Lemmas

## A.1 Notations

In this paper, $\sigma_i(M)$ denotes the $i$-th largest singular value of a matrix $M$, and $\lambda_i(M)$ denotes the $i$-th largest eigenvalue of $M$. $\|v\|$ denotes the Euclidean norm of a vector $v$, while $\|M\|_F$ and $\|M\|_2$ denote the Frobenius norm and induced $l_2$ norm of a matrix $M$, respectively. For a matrix $M$, $\mathrm{vec}(M)$ is the usual vectorization operation by stacking the columns of the matrix $M$ into a vector. For a vector $v \in \mathbb{R}^{n^2}$, $\mathrm{mat}(v)$ converts $v$ to a square matrix and $\mathrm{mat}_S(v)$ converts $v$ to a symmetric matrix, i.e., $\mathrm{mat}(v) = M$ and $\mathrm{mat}_S(v) = (M + M^T)/2$, where $M \in \mathbb{R}^{n \times n}$ is the unique matrix satisfying $v = \mathrm{vec}(M)$. $[n]$ denotes the integer set $[1, \ldots, n]$, and $\circ l$ stands for the shorthand of repeated cartesian product $\times \cdots \times$ for $l$ times. The symbol $\oslash$ denotes the kronecker product, while $\otimes$ denotes tensor outer product. $\asymp$ denotes "asymptotic to", meaning that the two terms on both sides of this symbol have the same order of magnitude.

## A.2 Critical Conditions for Unlifted Problem

We present the FOP and SOP conditions for the unlifted problem as our benchmark.

**Lemma 3.** *The vector $\hat{X} \in \mathbb{R}^{n \times r}$ is an SOP of* (2) *if and only if*

$$\nabla f(\hat{X}\hat{X}^\top)\hat{X} = 0, \tag{13}$$

$$2\langle \nabla f(\hat{X}\hat{X}^\top), UU^\top \rangle + [\nabla^2 f(\hat{X}\hat{X}^\top)](\hat{X}U^\top + U\hat{X}^\top, \hat{X}U^\top + U\hat{X}^\top) \geq 0 \quad \forall U \in \mathbb{R}^{n \times r} \tag{14}$$

*with* (13) *being the necessary and sufficient condition for $\hat{X}$ to be an FOP.*

A proof to the above lemma can be found in many matrix sensing literatures, including [16, 17, 42], etc.

## A.3 Additional Definitions

**Definition 5** (RIP, [2]). Given a natural number $p$, the linear map $\mathcal{A} : \mathbb{R}^{n \times n} \mapsto \mathbb{R}^m$ is said to satisfy $\delta_p$-RIP if there is a constant $\delta_p \in [0, 1)$ such that

$$(1 - \delta_p)\|M\|_F^2 \leq \|\mathcal{A}(M)\|^2 \leq (1 + \delta_p)\|M\|_F^2$$

holds for all matrices $M \in \mathbb{R}^{n \times n}$ satisfying $\mathrm{rank}(M) \leq p$.

**Definition 6** (Symmetric Tensor). Similar to the definition of symmetric matrices, for an order-$l$ tensor $\mathbf{a}$ with the same dimensions (i.e., $n_1 = \cdots = n_l$), also called a cubic tensor, it is said that the tensor is symmetric if its entries are invariance under any permutation of their indices:

$$a_{i_{\sigma(1)}\cdots i_{\sigma(l)}} = a_{i_1 \cdots i_l} \quad \forall \sigma, \quad i_1, \ldots, i_l \in \{1, \ldots, n\}$$

where $\sigma \in \mathcal{G}_l$ denotes a specific permutation and $\mathcal{G}_l$ is the symmetric group of permutations on $\{1, \ldots, l\}$. We denote the set of symmetric tensors as $\mathrm{S}^l(\mathbb{R}^n)$.

**Definition 7** (Rank of Tensors). The rank of a cubic tensor $\mathbf{a} \in \mathbb{R}^{n \circ l}$ is defined as

$$\mathrm{rank}(\mathbf{a}) = \min\{r | \mathbf{a} = \sum_{i=1}^{r} u_i \otimes v_i \otimes \cdots \otimes w_i\}$$

for some vector $u_i, \ldots, w_i \in \mathbb{R}^n$. Furthermore, according to [43], if $\mathbf{a}$ is a symmetric tensor, then it can be decomposed as:

$$\mathbf{a} = \sum_{i=1}^{r} \lambda_i u_i \otimes \cdots \otimes u_i := \sum_{i=1}^{r} \lambda_i u_i^{\otimes l}$$

and the rank is conveniently defined as the number of nonzero $\lambda_i$'s, which is very similar to the rank of symmetric matrices indeed. The most important concept in our paper is rank-1 tensors, and for any tensor $\mathbf{a}$, a necessary and sufficient condition for it to be rank-1 is that

$$\mathbf{a} = u^{\otimes l}$$

for some $u \in \mathbb{R}^n$.

**Definition 8** (Tensor Multiplication). Outer product is an operation carried out on a pair of tensors, denoted as $\otimes$. The outer product of 2 tensors $\mathbf{a}$ and $\mathbf{b}$, respectively of orders $l$ and $p$, is a tensor of order $l + p$, denoted as $\mathbf{c} = \mathbf{a} \otimes \mathbf{b}$ such that:

$$c_{i_1 \ldots i_l j_1 \ldots j_p} = a_{i_1 \ldots i_l} b_{j_1 \ldots j_p}$$

When the 2 tensors are of the same dimension, this product is such that $\otimes : \mathbb{R}^{n \circ l} \times \mathbb{R}^{n \circ p} \mapsto \mathbb{R}^{n \circ (l+p)}$. Henceforth, we use the shorthand notation

$$\underbrace{a \otimes \cdots \otimes a}_{l \text{ times}} := a^{\otimes l}$$

We also define an inner product of two tensors. The mode-$q$ inner product between the 2 aforementioned tensors having the same $q$-th dimension is denoted as $\langle \mathbf{a}, \mathbf{b} \rangle_q$. Without loss of generality, assume that $q = 1$ and

$$[\langle \mathbf{a}, \mathbf{b} \rangle_q]_{i_2 \ldots i_l j_2 \ldots j_p} = \sum_{\alpha=1}^{n_q} a_{\alpha i_2 \ldots i_l} b_{\alpha j_2 \ldots j_p}$$

Note that when we write $\langle \cdot, \cdot \rangle_q$, we count the $q$-th dimension of the first entry. Indeed, this definition of inner product can also be trivially extended to multi-mode inner products by just summing over all modes, denoted as $\langle \mathbf{a}, \mathbf{b} \rangle_{q,\ldots,s}$.

## A.4 Technical Lemmas

**Lemma 4** (Section 10.2 [44]). *For four arbitrary matrices $A, B, C, D$ of compatible dimensions, it holds that*

$$\langle A \otimes B, C \otimes D \rangle_{2,4} = AC \otimes BD \tag{15}$$

**Lemma 5** ([45]). *For any SOP $\hat{X}$ of (2), define $G$ as $G := -\lambda_{min}(\nabla f(\hat{X}\hat{X}^\top))$, and $L_s$ be the RSS constant. Then it holds that*
$$G \le \lambda_r(\hat{X}\hat{X}^\top) L_s$$
*where $r$ is the search rank of (2).*

**Lemma 6.** *Given an FOP $\hat{X}$ of (2), it holds that*

$$\lambda_r(\hat{X}\hat{X}^\top) < \sqrt{\frac{2L_s}{r\alpha_s}} \|M^*\|_F \tag{16}$$

*Proof of Lemma 6.* Lemma 6 of [17] states that given an arbitrary constant $\lambda$ and matrix $X \in \mathbb{R}^{n \times r}$, one can write

$$\|XX^\top\|_F^2 \ge \max\left\{ \frac{2L_s}{\alpha_s} \|M^*\|_F^2, \left(\frac{2\lambda\sqrt{r}}{\alpha_s}\right)^{4/3} \right\} \implies \|\nabla h(X)\|_F \ge \lambda$$

A simple negation to both sides gives

$$\|\nabla h(X)\|_F < \lambda \implies \|XX^\top\|_F^2 < \max\left\{ \frac{2L_s}{\alpha_s} \|M^*\|_F^2, \left(\frac{2\lambda\sqrt{r}}{\alpha_s}\right)^{4/3} \right\}$$

If we set $X = \hat{X}$, then left-hand side of the above inequality is automatically satisfied for small values of $\lambda$ since $\|\nabla h(\hat{X})\|_F = 0$, and thus we conclude that

$$\|\hat{X}\hat{X}^\top\|_F^2 < \frac{2L_s}{\alpha_s} \|M^*\|_F^2$$

since $\left(\frac{2\lambda\sqrt{r}}{\alpha_s}\right)^{4/3}$ can be made arbitrarily small. Therefore,

$$\|\hat{X}\hat{X}^\top\|_F^2 \ge r\lambda_r(\hat{X}\hat{X}^\top)^2 \implies \lambda_r(\hat{X}\hat{X}^\top) < \sqrt{\frac{2L_s}{r\alpha_s}} \|M^*\|_F$$

as $\hat{X}\hat{X}^\top$ can have at most $r$ eigenvalues due to its factorized form. $\qquad\square$

# B  Additional Details for Lifted Formulation of General $r$

We analyze (5) and generalize the results of [21] to $r > 1$. We start with the characterization of FOPs and SOPs of (5).

**Lemma 7.** *The tensor $\hat{\mathbf{w}} \in \mathbb{R}^{nr \circ l}$ is an SOP of (5) if and only if*

$$\langle \nabla f^l(\langle \mathbf{P}(\hat{\mathbf{w}}), \mathbf{P}(\hat{\mathbf{w}}) \rangle_{2*[l]}), \mathbf{P}(\hat{\mathbf{w}}) \rangle_{2*[l]} = 0, \tag{17a}$$

$$2\langle \nabla f^l(\langle \mathbf{P}(\hat{\mathbf{w}}), \mathbf{P}(\hat{\mathbf{w}}) \rangle_{2*[l]}), \langle \mathbf{P}(\Delta), \mathbf{P}(\Delta) \rangle_{2*[l]} +$$

$$\|\langle \mathbf{A}^{\otimes l}, \langle \mathbf{P}(\hat{\mathbf{w}}), \mathbf{P}(\Delta) \rangle_{2*[l]} + \langle \mathbf{P}(\Delta), \mathbf{P}(\hat{\mathbf{w}}) \rangle_{2*[l]} \rangle\|_F^2 \geq 0 \quad \forall \Delta \in \mathbb{R}^{nr \circ l} \tag{17b}$$

*with* (17b) *being a necessary and sufficient condition for $\hat{\mathbf{w}}$ to be a FOP.*

*Proof of Lemma 7.* We have

$$\nabla f^l(\mathbf{M}) = \langle \langle \mathbf{A}^{\otimes l}, \mathbf{M} - \mathcal{M}(\text{vec}(Z)^{\otimes l}) \rangle, \mathbf{A}^{\otimes l} \rangle_{1,4,\dots,3l-2} \tag{18}$$

where the new map $\mathcal{M} : \mathbb{R}^{nr \circ l} \mapsto \mathbb{R}^{n \circ 2l}$ is defined as

$$\mathcal{M}(\mathbf{w}) = \langle \mathbf{P}(\mathbf{w}), \mathbf{P}(\mathbf{w}) \rangle_{2*[l]},$$

and its total derivative at $\mathbf{w}$ is the linear map $D_{\mathbf{w}}\mathcal{M} : \mathbb{R}^{nr \circ l} \mapsto \mathbb{R}^{n \circ 2l}$ given below:

$$D_{\mathbf{w}}\mathcal{M}(\mathbf{v}) = \langle \mathbf{P}(\mathbf{v}), \mathbf{P}(\mathbf{w}) \rangle_{2*[l]} + \langle \mathbf{P}(\mathbf{w}), \mathbf{P}(\mathbf{v}) \rangle_{2*[l]}. \tag{19}$$

Combining (18) and (19) gives that

$$D_{\mathbf{w}}h^l(\mathbf{v}) = \langle \mathbf{A}^{\otimes l}, D_{\mathbf{w}}\mathcal{M}(\mathbf{v}) \rangle^\top \langle \mathbf{A}^{\otimes l}, \mathcal{M}(\mathbf{w}) - \mathcal{M}(\text{vec}(Z)^{\otimes l}) \rangle \tag{20}$$

The sensing matrices $A_k \; \forall k \in [m]$ are assumed to be symmetric, and therefore $\langle \mathbf{A}^{\otimes l}, D_{\mathbf{w}}\mathcal{M}(\mathbf{v}) \rangle = 2\langle \mathbf{A}^{\otimes l}, \langle \mathbf{P}(\mathbf{v}), \mathbf{P}(\mathbf{w}) \rangle_{2*[l]} \rangle$.

Therefore, since the first-order optimality condition for (5) is that $D_{\mathbf{w}}h^l(\mathbf{v}) = 0 \; \forall \mathbf{v} \in \mathbb{R}^{nr \circ l}$, it can be equivalently written as

$$\langle \langle \mathbf{A}^{\otimes l}, \mathbf{P}(\mathbf{w}) \rangle_{2*[l]}, \langle \mathbf{A}^{\otimes l}, \mathcal{M}(\mathbf{w}) - \mathcal{M}(\text{vec}(Z)^{\otimes l}) \rangle \rangle_{1,3,\dots,2l-1} = 0, \tag{21}$$

and left-hand side of the above equation yields (17a) after rearrangements.

For the second-order optimality condition, one can directly take the derivative of $D_{\mathbf{w}}h^l(\mathbf{v})$, but there is an easier way since we are only concerned the expression of its quadratic form evaluated at some tensor $\Delta \in \mathbb{R}^{nr \circ l}$. For a brief moment, assume that we aim to optimize over $\mathbf{X} \in \mathbb{R}^{[n \times r] \circ l}$, for which

$$\nabla h^l(\mathbf{X}) = 2\langle \nabla f^l(\langle \mathbf{X}, \mathbf{X} \rangle_{2*[l]}), \mathbf{X} \rangle_{2*[l]} \in \mathbb{R}^{[n \times r] \circ l}$$

Therefore, if we instead take the derivate of $g(\mathbf{P}(\mathbf{w}))$ with respect to $\mathbf{w}$, we can simply use the chain rule and arrive at

$$\nabla_{\mathbf{w}} h^l(\mathbf{P}(\mathbf{w})) = \langle \nabla h^l(\mathbf{X}), \mathbf{P}^{\otimes l} \rangle_{1,2,4,5,\dots,3l-1,3l} \tag{22}$$

Hence, if we take the derivate of $\nabla h^l$ and evaluate it at $\mathbf{X}$ in the direction of $\mathbf{U} \in \mathbb{R}^{[n \times r] \circ l}$, we obtain that

$$D_{\mathbf{X}}\nabla h^l(\mathbf{U}) = 2\langle \nabla f^l(\langle \mathbf{X}, \mathbf{X} \rangle_{2*[l]}), \mathbf{U} \rangle_{2*[l]} + \langle \langle \mathbf{A}^{\otimes l}, \langle \mathbf{X}, \mathbf{U} \rangle_{2*[l]} + \langle \mathbf{U}, \mathbf{X} \rangle_{2*[l]} \rangle, \langle \mathbf{A}^{\otimes l}, \mathbf{w} \rangle_{2,5,\dots,3l-1} \rangle$$

$$+ \langle \langle \mathbf{A}^{\otimes l}, \langle \mathbf{X}, \mathbf{U} \rangle_{2*[l]} + \langle \mathbf{U}, \mathbf{X} \rangle_{2*[l]} \rangle, \langle \mathbf{A}^{\otimes l}, \mathbf{w} \rangle_{3,6,\dots,3l} \rangle$$

Combined with (22), we conclude that

$$[\nabla_{\mathbf{w}}^2 h^l(\mathbf{P}(\mathbf{w}))](\mathbf{v}, \mathbf{v}) = 2\langle \nabla f^l(\mathcal{M}(\mathbf{w})), \mathcal{M}(\mathbf{v}) \rangle + \langle \langle \mathbf{A}^{\otimes l}, D_{\mathbf{w}}\mathcal{M}(\mathbf{v}) \rangle, \langle \mathbf{A}^{\otimes l}, D_{\mathbf{w}}\mathcal{M}(\mathbf{v}) \rangle \rangle$$

$$= 2\langle \nabla f^l(\mathcal{M}(\mathbf{w})), \mathcal{M}(\mathbf{v}) \rangle + \|\langle \mathbf{A}^{\otimes l}, D_{\mathbf{w}}\mathcal{M}(\mathbf{v}) \rangle\|_F^2$$

which yields (17b) directly. $\qquad\square$

Now, we turn to showcasing the relationship between the FOPs of (5) and those of (2), which also have a one-to-one correspondence in the symmetric rank-1 regime. This is the reason why it is necessary to introduce (5) despite the extra complication, as rank-1 components tensors in $\mathbb{R}^{[n \times r] \circ l}$ are not lifted versions of $X \in \mathbb{R}^{n \times r}$.

**Theorem 3.** *For the lifted formulation* (5)*, the first-order condition* $\nabla h^l(\hat{\mathbf{w}}) = 0$ *holds for a symmetric rank-1 tensor* $\hat{\mathbf{w}}$ *if and only if*

$$\hat{\mathbf{w}} = \mathrm{vec}(\hat{X})^{\otimes l}$$

*where* $\hat{X} \in \mathbb{R}^{n \times r}$ *is an FOP of* (2)*.*

*Proof of Theorem 3.* When $\hat{\mathbf{w}} = \mathrm{vec}(\hat{X})^{\otimes l}$, Lemma A.4 and (17a) together imply that

$$\langle \nabla f^l(\langle \hat{X}^{\otimes l}, \hat{X}^{\otimes l} \rangle_{2*[l]}), \hat{X}^{\otimes l} \rangle_{2*[l]} = (\nabla f(\hat{X}\hat{X}^\top)\hat{X})^{\otimes l} = 0 \tag{23}$$

which is equivalent to

$$\nabla f(\hat{X}\hat{X}^\top)\hat{X} = 0,$$

which is exactly (13). $\qquad\square$

Theorem 3 establishes a robust connection between the first-order critical points of the lifted formulation and those of the unlifted formulation. This implies that when first-order methods approach a critical point in (5), valuable information about an FOP of (2) can also be readily extracted. However, the primary challenge in optimizing (2) stems from spurious solutions, which cannot be escaped by first or even second-order algorithms. Consequently, it becomes crucial to examine whether the Hessians of the FOPs of (5), especially those that correspond to the spurious solutions of (2), exhibit any unique properties. As it turns out, the non-global FOPs of (5) display some highly favorable characteristics: they no longer constitute second-order critical points of (5) and are transformed into strict saddles when the parametrization level $l$ is sufficiently large.

To motivate our analysis of conversion from spurious solutions to strict saddle points, we first offer a closer analysis to the SOPs of the unlifted problem (2), which also serves as the key intuition into our main results in this section.

The main observation is that, for a spurious SOP $\hat{X}$ and any ground truth $Z$ with $\hat{X}\hat{X}^\top \neq ZZ^\top$, although they all obey conditions (13) and (14), they still have intrinsic differences that can be amplified via over-parametrization. To illustrate this phenomenon in more detail, we will introduce the following Lemma:

**Lemma 8.** *For an arbitrary FOP* $\hat{X} \in \mathbb{R}^{n \times r}$ *of* (2) *satisfying the* $(\alpha_s, r)$-*RSC property, the following inequality holds:*

$$\lambda_{\min}(\nabla f(\hat{X}\hat{X}^\top)) \leq -\alpha_s \frac{\|\hat{X}\hat{X}^\top - M^*\|_F^2}{2\,\mathrm{tr}(M^*)} \leq 0 \tag{24}$$

*Proof for Lemma 8.* According to [17], $\nabla f(M)$ can be assumed to be symmetric without loss of generality. Hence, one can select $u \in \mathbb{R}^n$ such that $u^\top \nabla f(\hat{x}\hat{x}^\top)u = \lambda_{\min}(\nabla f(\hat{x}\hat{x}^\top))$. Then via the definition of RSC we have

$$f(M^*) \geq f(\hat{X}\hat{X}^\top) + \langle \nabla f(\hat{X}\hat{X}^\top), M^* - \hat{X}\hat{X}^\top \rangle + \frac{\alpha_s}{2}\|\hat{X}\hat{X}^\top - M^*\|_F^2.$$

Given that $\hat{X}$ is also an FOP, we have that

$$\langle \nabla f(\hat{X}\hat{X}^\top), \hat{X}\hat{X}^\top \rangle = 0$$

according to (13) and since $f(\hat{X}\hat{X}^\top) - f(M^*) \geq 0$, one can write that

$$\langle \nabla f(\hat{X}\hat{X}^\top), M^* \rangle \leq -\frac{\alpha_s}{2}\|\hat{x}\hat{x}^\top - M^*\|_F^2$$

after rearrangements. Furthermore, since both $\nabla f(\hat{X}\hat{X}^\top)$ and $M^*$ are assumed to be positive semidefinite for the above-mentioned reasons, we have that

$$\langle \nabla f(\hat{X}\hat{X}^\top), M^* \rangle \geq \lambda_{\min}(\nabla f(\hat{X}\hat{X}^\top))\,\mathrm{tr}(M^*)$$

which implies that

$$\lambda_{\min}(\nabla f(\hat{X}\hat{X}^\top)) \leq -\alpha_s \frac{\|\hat{X}\hat{X}^\top - M^*\|_F^2}{2\,\mathrm{tr}(M^*)} \leq 0 \tag{25}$$

This completes the proof. $\qquad\square$

Now let us recall (14), which can be stated equivalently as

$$\lambda_{\min}(\nabla f(\hat{X}\hat{X}^\top)) \geq -[\nabla^2 f(\hat{X}\hat{X}^\top)](\hat{X}U^\top + U\hat{X}^\top, \hat{X}U^\top + U\hat{X}^\top) \quad \forall U$$

By using the $(L_s, r)$-RSS property and the assumption that the sensing matrices are symmetric, we can further lower-bound the right-hand side of the above inequality as

$$-[\nabla^2 f(\hat{X}\hat{X}^\top)](\hat{X}U^\top + U\hat{X}^\top, \hat{X}U^\top + U\hat{X}^\top) \geq -4[\nabla^2 f(\hat{X}\hat{X}^\top)](\hat{X}U^\top) \geq -4L_s\|\hat{X}U^\top\|_F^2$$

Therefore, it is easy to see that a sufficient condition for the spurious SOPs to disappear is

$$\alpha_s \frac{\|\hat{X}\hat{X}^\top - M^*\|_F^2}{2\operatorname{tr}(M^*)} \geq 4L_s\|\hat{X}U^\top\|_F^2 \quad \forall U \tag{26}$$

which means that the $L_s$ and $\alpha_s$ parameters should be benign, and this essentially constitutes the main proof strategy in the existing literature showing in-existence of spurious solutions under benign RIP or RSS/RSC conditions [14, 17, 46, 15, 45].

Therefore, it is natural to ask, in the case when $L_s$ and $\alpha_s$ do not satisfy (26), whether one can systematically over-parametrize the problem so that the LHS of (26) eventually becomes bigger than the RHS. We know that if we just raise both the RHS and LHS to arbitrary powers, the sign of the inequality will not flip. Therefore, the key insight is that if we keep the constant 4 unchanged, and lift the other terms to arbitrary powers, we can eventually satisfy (14). In general terms, we take the following steps in order to establish a strong result regarding the conversion of spurious solutions to strict saddle points:

1. Proving that $\langle \nabla f^l(\langle \mathbf{P}(\hat{\mathbf{w}}), \mathbf{P}(\hat{\mathbf{w}}) \rangle, \Delta \otimes \Delta \rangle \geq |\lambda_{\min}(\nabla f(\hat{X}\hat{X}^\top))|^l$ for some appropriately chosen point $\Delta \in \mathbb{R}^{nr \circ l}$.

2. Proving that $\|\langle \mathbf{A}^{\otimes l}, \langle \mathbf{P}(\mathbf{w}), \mathbf{P}(\Delta) \rangle_{2*[l]} + \langle \mathbf{P}(\Delta), \mathbf{P}(\mathbf{w}) \rangle_{2*[l]} \rangle\|_F^2 \leq 4L_s\|\hat{X}U^\top\|_F^{2l}$ for some appropriately chosen points $\Delta \in \mathbb{R}^{nr \circ l}$ and $U \in \mathbb{R}^{n \times r}$

3. Finding the smallest $l$ that converts the spurious solution to strict saddle point, under mild technical conditions.

Now we turn to the main result of the general-rank scenario, which concerns the conversion of spurious solutions to strict saddle points. We present the formal results below.

**Theorem 4.** *Consider an SOP $\hat{X} \in \mathbb{R}^{n \times r}$ of (2) of general rank $r < n$, such that $\hat{X}\hat{X}^\top \neq M^*$, and assume that (2) satisfies the RSC and RSS conditions. Then $\hat{\mathbf{w}} = \operatorname{vec}(\hat{X})^{\otimes l}$ is a strict saddle of (5) with a rank-1 symmetric escape direction if $\hat{X}$ satisfies the inequality*

$$\|M^* - \hat{X}\hat{X}^\top\|_F^2 \geq \frac{L_s}{\alpha_s} \lambda_r(\hat{X}\hat{X}^\top) \operatorname{tr}(M^*) \tag{27}$$

*and $l$ is odd and is large enough so that*

$$l > \frac{1}{1 - \log_2(2\beta)} \tag{28}$$

*where $\beta$ is defined as*

$$\beta := \frac{L_s \operatorname{tr}(M^*) \lambda_r(\hat{X}\hat{X}^\top)}{\alpha_s \|M^* - \hat{X}\hat{X}^\top\|_F^2}.$$

*Here, $L_s$ and $\alpha_s$ are the respective RSS and RSC constants of (2).*

*Proof of Theorem 4.* By Lemma 8, we select $u \in \mathbb{R}^n$ such that $u^\top \nabla f(\hat{X}\hat{X}^\top)u = \lambda_{\min}(\nabla f(\hat{X}\hat{X}^\top))$ with $\lambda_{\min}(\nabla f(\hat{X}\hat{X}^\top)) \leq 0$.

Now define $G := -\lambda_{\min}(\nabla f(\hat{X}\hat{X}^\top)) \geq 0$. If we label

$$C_1 := \langle \nabla f(\hat{X}\hat{X}^\top), UU^\top \rangle, \quad C_2 := [\nabla^2 f(\hat{X}\hat{X}^\top)](\hat{X}U^\top, \hat{X}U^\top)$$

Then we have that $C_1 = -G$. Also, since the sensing matrices $A_a$ can be assumed be to symmetric, we have that

$$[\nabla^2 f(\hat{X}\hat{X}^\top)](\hat{X}U^\top + U\hat{X}^\top, \hat{X}U^\top + U\hat{X}^\top) = 4[\nabla^2 f(\hat{X}\hat{X}^\top)](\hat{X}U^\top, \hat{X}U^\top).$$

Additionally we choose $q \in \mathbb{R}^r$ to be the $r$-th singular value of $\hat{X}$, with

$$\|\hat{X}q\|_2 = \sigma_r(\hat{X}), \qquad \|q\|_2 = 1$$

and define $U \in \mathbb{R}^{n \times r} = uq^\top$. Subsequently, the RSS condition can be used to show that

$$[\nabla^2 f(\hat{X}\hat{X}^\top)](\hat{X}U^\top + U\hat{X}^\top, \hat{X}U^\top + U\hat{X}^\top) \leq L_s \|\hat{X}U^\top + U\hat{X}^\top\|_F^2$$
$$= L_s \|u(\hat{X}q)^\top + (\hat{X}q)u^\top\|_F^2 = 2L_s\|\hat{X}q\|_F^2 + 2L_s(q^\top(\hat{X}^\top u))^2 = 2L_s\lambda_r(\hat{X}\hat{X}^\top)$$

since $\hat{X}^\top u = 0$ according to the first-order condition (13). Therefore,

$$C_2 \leq \frac{1}{2}L_s\lambda_r(\hat{X}\hat{X}^\top)$$

Now, if we choose $\Delta = \text{vec}(U)^{\otimes l}$ for the aforementioned $U \in \mathbb{R}^{n \times r}$, the LHS of (17b) can be expressed as:

$$\text{LHS} = 2(\langle \mathbf{A}, \hat{X}\hat{X}^\top \rangle_{2,3}^\top \langle \mathbf{A}, uu^\top \rangle_{2,3})^l - 2(\langle \mathbf{A}, M^* \rangle_{2,3}^\top \langle \mathbf{A}, uu^\top \rangle_{2,3})^l + 4(\|\langle \mathbf{A}, \hat{X}U^\top \rangle_{2,3}\|_2^2)^l$$
$$\leq 2(\lambda_{\min}(\nabla f(\hat{X}\hat{X}^\top)))^l + 4C_2^l$$
$$= 2C_1^l + 4C_2^l \tag{29}$$

where the inequality follows from:

$$a^n - b^n \leq (a-b)^n, \quad \forall b \geq a \geq 0$$

Here, since $a - b = C_1 \leq 0$, the above inequality can be used. As a result,

$$\text{LHS of (17b)} \leq \underbrace{-2G^l}_{\text{Part 1}} + \underbrace{\frac{2}{2^{l-1}}L_s^l\lambda_r(\hat{X}\hat{X}^\top)^l}_{\text{Part 2}}$$

We know since $G \geq 0$, Part 1 is always negative assuming $l$ is odd, and Part 2 is always positive. Therefore, it suffices to find an order $l$ such that

$$G^l > (1/2^{l-1})L_s^l\lambda_r(\hat{X}\hat{X}^\top)^l \tag{30}$$

To derive a sufficient condition for (30), we first need a lower bound on $G$, and Lemma (8) conveniently provides this bound, giving that

$$G \geq \frac{\alpha_s}{2\,\text{tr}(M^*)}\|M^* - \hat{X}\hat{X}^\top\|_F^2 \tag{31}$$

Therefore, if

$$\left(\frac{\alpha_s}{2\,\text{tr}(M^*)}\|M^* - \hat{X}\hat{X}^\top\|_F^2\right)^l > (1/2^{l-1})L_s^l\lambda_r(\hat{X}\hat{X}^\top)^l,$$

we can conclude that (30) holds, which implies that the LHS of (17b) is negative, directly proving that $\hat{X}^{\otimes l}$ is not an SOP anymore. Elementary manipulations of the above equation give that a sufficient condition is

$$\|M^* - \hat{X}\hat{X}^\top\|_F^2 > 2^{1/l}\frac{L_s}{\alpha_s}\lambda_r(\hat{X}\hat{X}^\top)\,\text{tr}(M^*) \tag{32}$$

We now consider (27), which means that

$$\lambda_r(\hat{X}\hat{X}^\top) \leq \frac{\alpha_s}{L_s\,\text{tr}(M^*)}\|M^* - \hat{X}\hat{X}^\top\|_F^2 \tag{33}$$

Subsequently, define a constant $\gamma$ such that

$$L_s\lambda_r(\hat{X}\hat{X}^\top) = \gamma\left(\frac{\alpha_s}{2\,\text{tr}(M^*)}\|M^* - \hat{X}\hat{X}^\top\|_F^2\right)$$

Then, according to Lemma 5 and (31), we can conclude that $\gamma \geq 1$. Moreover, (33) also means that $\gamma < 2$. With this new definition, the sufficient condition (32) becomes

$$1 > \frac{\gamma}{2^{(l-1)/l}} \tag{34}$$

Since we already know that $1 \le \gamma < 2$, there always exists a large enough $l$ such that (34) holds, which in turn implies that LHS of (17b) is negative, proving that $\text{vec}(\hat{X})^{\otimes l}$ is a saddle point with the escape direction $\text{vec}(U)^{\otimes l}$, proving the claim.

Next, we aim to study how large $l$ needs to be in order for (34) to hold. Again, we know that

$$\gamma = \frac{2L_s \, \text{tr}(M^*) \lambda_r(\hat{X}\hat{X}^\top)}{\alpha_s \|M^* - \hat{X}\hat{X}^\top\|_F^2} := 2\beta$$

and that $\beta \le 1$ due to assumption (27). Therefore, for (34) to hold true, it is enough to have

$$2^{(l-1)/l} > 2\beta \implies \frac{l-1}{l} > \log_2(2\beta) \implies l > \frac{1}{1 - \log_2(2\beta)}$$

$\square$

## B.1 Other Considerations of Lifted Landscape

In the previous sections, we have shown that by lifting the optimization problem (2) into tensor spaces, we could convert spurious local solutions into strict saddle points. However, it is also important that we could distinguish the true ground truth solutions $Z \in \mathbb{R}^{n \times r}$ with $ZZ^\top \doteq M^*$ from the spurious ones. This requires that the true solutions $Z$ will remain SOPs after lifting, which we indeed prove in the following theorem:

**Theorem 5.** *Assume that $Z \in \mathbb{R}^{n \times r}$ is a ground truth solution of* (2) *such that $ZZ^\top = M^*$. Then* $\text{vec}(Z)^{\otimes l}$ *remains an SOP of* (5) *regardless of the parametrization level $l$, and without the need for* (2) *to satisfy the RSC or RSS conditions.*

*Proof of Theorem 5.* Let us start with the first-order optimality condition. Consider the linear map in the proof of Lemma 7 $\mathcal{M} : \mathbb{R}^{nrol} \mapsto \mathbb{R}^{no2l}$

$$\mathcal{M}(\mathbf{w}) = \langle \mathbf{P}(\mathbf{w}), \mathbf{P}(\mathbf{w}) \rangle_{2*[l]},$$

Again, it is apparent that

$$\nabla f^l(\mathbf{M}) = \langle \langle \mathbf{A}^{\otimes l}, \mathbf{M} - \mathcal{M}(\text{vec}(Z)^{\otimes l}) \rangle, \mathbf{A}^{\otimes l} \rangle_{1,4,\dots,3l-2}$$

Therefore, at the point $\mathbf{M} = \mathcal{M}(\text{vec}(Z)^{\otimes l})$, we know that $\nabla f^l(\mathcal{M}(\text{vec}(Z)^{\otimes l})) = 0$. Consequently, the LHS of (17a) is equal to zero since it is a product between $\nabla f^l(\mathcal{M}(\text{vec}(Z)^{\otimes l}))$ and $\mathbf{P}(\text{vec}(Z)^{\otimes l})$.

Next, we turn to the second-order optimality condition. Again, recall from the proof of Lemma 7 that

$$\text{LHS of (17b)} = \underbrace{2\langle \nabla f^l(\mathcal{M}(\mathbf{w})), \mathcal{M}(\mathbf{\Delta}) \rangle}_{\text{Part 1}} + \underbrace{\|\langle \mathbf{A}^{\otimes l}, D_\mathbf{w}\mathcal{M}(\mathbf{\Delta}) \rangle\|_F^2}_{\text{Part 2}}$$

By the above arguments, we have $\nabla f^l(\mathcal{M}(\mathbf{w})) = 0$ when $\mathbf{w} = \text{vec}(Z)^{\otimes l}$, meaning that Part 1 equals to zero. This implies that

$$\text{LHS of (17b)} = \|\langle \mathbf{A}^{\otimes l}, D_\mathbf{w}\mathcal{M}(\mathbf{\Delta}) \rangle\|_F^2 \ge 0, \qquad \forall \mathbf{\Delta}$$

regardless of the values of $\mathbf{A}$ or $\mathbf{w} = \text{vec}(Z)^{\otimes l}$. $\square$

Next, it is important to analyze the main results obtained in Theorem 4 under the lens of some other existing characterizations of the loss landscape of (2). According to Theorem 4, over-parametrization or lifting proves to be highly beneficial when dealing with spurious solutions, represented by $\hat{X}$, that significantly deviate from the actual ground truth. The theorem implies that as the distance between $\hat{X}$ and the ground truth increases, a smaller value of $l$ is necessary for $\text{vec}(\hat{X})^{\otimes l}$ to evolve into a saddle point, as alluded to in (28). This concept is consistent with previous studies, which maintain that the area surrounding $M^*$ exhibits a favorable optimization landscape, characterized by an absence of deceptive local solutions in a specified zone around $M^*$. A commonly cited illustration of this assertion is provided below.

**Theorem 6** (Theorem 3 [47])**.** *If $\hat{X}$ is an SOP of* (2) *and*

$$\|\hat{X}\hat{X}^\top - M^*\|_F \leq \frac{4L_s\alpha_s}{(L_s + \alpha_s)^2}\lambda_r(M^*),\tag{35}$$

*then*

$$\hat{X}\hat{X}^\top = M^*$$

This means that any spurious solution of (2) is reasonably far away from the ground truth solution $M^*$. Coupled with the fact that lifting the problem into higher-dimensional tensor spaces can convert spurious solutions far away from $M^*$ into strict saddles points, we can ascertain that by setting the RHS of (35) to be greater or equal to the RHS of (27), all spurious solutions will be converted into strict saddle points via lifting. We make this key observation concrete in the following theorem.

**Theorem 7.** *Assume that $\hat{X} \in \mathbb{R}^{n \times r}$ is a spurious solution of* (2)*, and that* (2) *satisfies the RSC and RSS assumptions with the $\alpha_s$ and $L_s$ constants, respectively. Then* $\mathrm{vec}(\hat{X})^{\otimes l}$ *is a strict saddle point of* (5) *for an odd $l$ satisfying* (28) *if*

$$\|M^*\|_F \leq \frac{1}{\tau\sqrt{r}}\frac{2\sqrt{2}\alpha_s^{5/2}}{(L_s + \alpha_s)^2\sqrt{L_s}}\tag{36}$$

*where $\tau$ is the condition number of $M^*$.*

*Proof of Theorem 7.* By using Lemma 6, we know that

$$\text{RHS of (27)} \leq \sqrt{\frac{2L_s^3}{r\alpha_s^3}}\|M^*\|_F\,\mathrm{tr}(M^*)$$

Hence, it is enough to make the RHS of the above inequality to be less than that of (35), meaning that

$$\sqrt{\frac{2L_s^3}{r\alpha_s^3}}\|M^*\|_F\,\mathrm{tr}(M^*) \leq \frac{4L_s\alpha_s}{(L_s + \alpha_s)^2}\lambda_r(M^*) \implies \|M^*\|_F\frac{\mathrm{tr}(M^*)}{\lambda_r(M^*)} \leq \frac{2\sqrt{2r}\alpha_s^{5/2}}{(L_s + \alpha_s)^2\sqrt{L_s}}$$

Then, acknowledging that $\mathrm{tr}(M^*) \leq r\tau\lambda_r(M^*)$ completes the proof. $\qquad\square$

# C Additional Details for Implicit Bias of GD in Tensor Space

## C.1 More Tensor Algebra

**Definition 9.** Given a cubic tensor $\mathbf{w} \in \mathbb{R}^{n \circ l}$, its spectral norm $\| \cdot \|_S$ and nuclear norm $\| \cdot \|_*$ are defined respectively as

$$\|\mathbf{w}\|_* = \inf \left\{ \sum_{j=1}^{r_m} |\lambda_j| : \mathbf{w} = \sum_{j=1}^{r_m} \lambda_j w_j^{\otimes l}, \ \|w_j\|_2 = 1, w_j \in \mathbb{R}^n \right\}$$

$$\|\mathbf{w}\|_S = \sup \left\{ |\langle \mathbf{w}, u^{\otimes l} \rangle| \ \|u\|_2 = 1, u \in \mathbb{R}^n \right\}$$

From the definition, it also follows that

$$\|\mathbf{w}\|_S \le \|\mathbf{w}\|_*$$

The above definitions are similar to those for their matrix counterparts. However, unlike the spectral norm of matrices, the spectral norm of tensors are not tensor norms, namely that they do not obey

$$\|\langle \mathbf{w}, \mathbf{v} \rangle\|_S \le \|\mathbf{w}\|_S \|\mathbf{v}\|_S$$

in general. Conversely, the nuclear norm is a valid tensor norm, and we have the following property:

**Lemma 9** (Theorem 2.1, 3.2 [38]). *For tensors $\mathbf{w}$ and $\mathbf{v}$ of appropriate dimensions (if doing inner product, the dimensions along which the multiplication is performed must have matching size), we have*

$$\|\langle \mathbf{w}, \mathbf{v} \rangle\|_S \le \|\mathbf{w}\|_S \|\mathbf{v}\|_*$$
$$\|\langle \mathbf{w}, \mathbf{v} \rangle\|_* \le \|\mathbf{w}\|_* \|\mathbf{v}\|_*$$

Moreover, they have a dual norm relationship:

**Lemma 10** (Lemma 21 [48]). *The spectral norm $\| \cdot \|_S$ is the dual norm to the nuclear norm $\| \cdot \|_*$, namely given an arbitrary tensor $\mathbf{w}$, we have that*

$$\|\mathbf{w}\|_S = \sup_{\|\mathbf{v}\|_* \le 1} |\langle \mathbf{w}, \mathbf{v} \rangle|$$

*with $\mathbf{v}$ having the same dimensions as $\mathbf{w}$.*

Next, we introduce the notion of eigenvalues for tensors. There are many related definitions, like outlined in [39]. However, we introduce a novel variational characterization of eigenvalues that resembles the Courant-Fisher minimax definition for eigenvalues of matrices. Note this is a new definition that is fist introduced in this paper, and may be of independent interest outside of the current scope.

**Definition** (Definition 4, Variational Eigenvalue of Tensors). For a given tensor $\mathbf{w} \in \mathbb{R}^{n \circ l}$, we define its $k^{th}$ variational eigenvalue (v-Eigenvalue) $\lambda_k^v(\mathbf{w})$ as

$$\lambda_k^v(\mathbf{w}) := \max_{\substack{S \\ \dim(S)=k}} \min_{\mathbf{u} \in S} \frac{|\langle \mathbf{w}, \mathbf{u} \rangle|}{\|\mathbf{u}\|_F^2}, \quad k \in [n]$$

where $S$ is a subspace of $\mathbb{R}^{n \circ l}$ that is spanned by a set of orthogonal, symmetric, rank-1 tensors. Its dimension denotes the number of orthogonal tensors that span this space.

It is apparent from the definition that $\|\mathbf{w}\|_S = \lambda_1^v(\mathbf{w})$. Note that our definition of v-Eigenvalues of tensors can only define $n$ eigenvalues at most, which is not the maximum amount of H- or Z-Eigenvalues a tensor can have [39], and it is well known that even with symmetric tensors, its rank can go well beyond $n$ [49]. We also note that this definition exactly coincides with the definition of Hermitian tensor eigenvalues (introduced here [50]) when constrained to Hermitian tensors [51]. We also conjecture that this definition coincides with the top-n Z-Eigenvalues for even-order symmetric real tensors [39], but it is an open question for now.

Using the definition of v-Eigenvalues, we can also obtain an equivalent characterization, just like the Courant-Fisher definition for matrix eigenvalues, which helps us in proving a tensor version of Weyl's inequality:

**Proposition 3.** *For an integer $k$ in $[1, \ldots, n]$, the $k^{th}$ variational eigenvalue (v-Eigenvalue) $\lambda_k^v(\mathbf{w})$ of a tensor $\mathbf{w}$ satisfies:*

$$\lambda_k^v(\mathbf{w}) = \min_{\substack{T \\ \dim(T)=n-k+1}} \max_{\mathbf{u} \in T} \frac{|\langle \mathbf{w}, \mathbf{u} \rangle|}{\|\mathbf{u}\|_F^2} = \max_{\substack{S \\ \dim(S)=k}} \min_{\mathbf{u} \in S} \frac{|\langle \mathbf{w}, \mathbf{u} \rangle|}{\|\mathbf{u}\|_F^2}$$

*Proof of Proposition 3.* We prove the proposition by contradiction. Assume that the two formulations claimed to be identical in Proposition 3 are not the same. We further assume that $S$ is spanned by symmetric, rank-1 tensors $\{\mathbf{u}_1, \ldots, \mathbf{u}_k\}$, and that $T$ is spanned by symmetric, rank-1 tensors $\{\mathbf{u}_{-(n-k+1)}, \ldots, \mathbf{u}_{-1}\}$, meaning that

$$\langle \mathbf{w}, \mathbf{u}_k \rangle \neq \langle \mathbf{w}, \mathbf{u}_{-(n-k+1)} \rangle$$

assuming that $\mathbf{u}_k$ and $\mathbf{u}_{-(n-k+1)}$ are the *inner* argmin and argmax of their respective formulations with norm 1. Since they have to be rank-1 tensors (if not we can decrease the proportion of orthogonal elements with higher or lower $|\langle \mathbf{w}, \mathbf{u} \rangle|$ values), it is possible to denote

$$\mathbf{u}_k = u_k^{\otimes l}, \quad \mathbf{u}_{-(n-k+1)} = u_{-(n-k+1)}^{\otimes l} \text{ where } u_k, u_{-(n-k+1)} \in \mathbb{R}^n$$

We also know that $u_k$ and $u_{-(n-k+1)}$ are linearly independent, as otherwise $\mathbf{u}_k$ and $\mathbf{u}_{-(n-k+1)}$ will have the same inner product with $\mathbf{w}$. Thus, assume

$$u_k = \xi_1 u_{-(n-k+1)} + \xi_2 u_{-(n-k+1)}^\perp, \quad \xi_2 \neq 0.$$

It follows that

$$\mathbf{u}_k = \xi_1^l u_{-(n-k+1)}^{\otimes l} + \xi_2^l (u_{-(n-k+1)}^\perp)^{\otimes l} + \underbrace{\phantom{xxxxxx}}_{\text{other non-symmetric terms}}$$

Denote $(u_{-(n-k+1)}^\perp)^{\otimes l} := \mathbf{u}_{k+1}$. Now, it follows from definition that

$$\mathbf{u}_{k+1} \perp \{\mathbf{u}_1, \ldots, \mathbf{u}_{k-1}\}$$

and also

$$\mathbf{u}_{k+1} \notin \text{span}\{\mathbf{u}_{-(n-k)}, \ldots, \mathbf{u}_{-1}\}$$

as otherwise the *outer* maximization formulation affecting the choice of $u_k$ will make $\xi_2 = 0$, contradicting our claim. By definition we have $\text{span}\{\mathbf{u}_1, \ldots, \mathbf{u}_k\} \bigcap \text{span}\{\mathbf{u}_{-(n-k)}, \ldots, \mathbf{u}_{-1}\} = \{\emptyset\}$.

In summary we have that $\mathbf{u}_{k+1} \perp \mathbf{u}_{-(n-k+1)}, \{\mathbf{u}_1, \ldots, \mathbf{u}_{k-1}\}, \{\mathbf{u}_{-(n-k)}, \ldots, \mathbf{u}_{-1}\}$, meaning that we have obtained $n + 1$ symmetric rank-1 and $n$-dimensional tensors all orthogonal to each other, which is apparently not possible, thus refuting our initial claim. $\square$

With this new definition equipped, we proceed to show a tensor version of Weyl's inequality, which is key in our proof as promised.

**Lemma 11** (Tensor Weyl's). *Consider two tensors $\mathbf{w}$ and $\mathbf{v}$ of the same dimension. It holds that*

$$\lambda_k^v(\mathbf{w}) + \lambda_1^v(\mathbf{v}) \geq \lambda_k^v(\mathbf{w} + \mathbf{v}) \geq \lambda_k^v(\mathbf{w}) - \lambda_1^v(\mathbf{v}) \tag{37}$$

The proof of Lemma 11 is highly similar to that of Theorem 2 in [51], only substituting for our new definition of v-Eigenvalues, thus omitted for simplicity.

## C.2 Main Results and Their Proofs

*Note that in this section some tensor inner products will be written as if they were matrices for clarity of writing, and some subscripts for inner-products will be dropped when obvious. If two tensors in $\mathbb{R}^{n r \circ 2l}$ are multiplied together, then the even dimensions of the first tensor will be inner-producted with the odd dimensions of the second tensor. When a tensor in $\mathbb{R}^{n r \circ 2l}$ multiplies with a tensor in $\mathbb{R}^{n r \circ l}$, then the even dimensions of the first tensor will be inner-producted with all the dimensions of the second tensor.*

We start with the proof to Lemma 1.

*Proof of Lemma 1.* We proceed with the proof by induction. First, assume that $\mathbf{w}_0 = x_0^{\otimes l}$ for some $x_0 \in \mathbb{R}^{nr}$. One can write

$$\nabla h^l(\mathbf{w}_0) = \langle\langle(I_r \oslash_{1,2} \mathbf{A})^{\otimes l}, \mathbf{w}_0\rangle_{2*[l]}, \langle\mathbf{A}^{\otimes l}, \mathcal{M}(\mathbf{w}_0) - \mathcal{M}(\text{vec}(Z)^{\otimes l})\rangle\rangle_{1,3,\dots,2l-1} \qquad (38)$$

where $\mathcal{M}(\cdot)$ is defined per proof of Lemma 7. The difference between this formulation and (21) is that we have replaced $\langle\mathbf{A}^{\otimes l}, \mathbf{P}(\mathbf{w}_0)\rangle_{2*[l]}$ with $\langle(I_r \oslash_{1,2} \mathbf{A})^{\otimes l}, \mathbf{w}_0\rangle_{2*[l]}$, which are equivalent, just with the second tensor having the dimensions $nr, m, \dots, nr, m$ so that $\nabla h^l(\mathbf{w}_0)$ has the dimensions $nr, \dots, nr$. Note that $\oslash$ denotes the usual kronecker product, which can be thought of a reshaped version of tensor outer product. $\oslash_{1,2}$ denotes the kronecker product only happening with respect to the first 2 dimensions of $\mathbf{A}$. From now on, we denote $\mathbf{A}_r := I_r \oslash_{1,2} \mathbf{A}$.

Now, according to the above formulation and Lemma A.4, we have

$$\nabla h^l(\mathbf{w}_0) = \left(\langle\mathbf{A}_r, \langle\mathbf{A}, \text{mat}(x_0)\,\text{mat}(x_0)^\top - M^*\rangle\rangle_{3,6,\dots,3l}\, x_0\right)^{\otimes l}$$
$$:= (\langle\mathbf{A}_r^*\mathbf{A}, \text{mat}(x_0)\,\text{mat}(x_0)^\top - M^*\rangle\, x_0)^{\otimes l} \qquad (39)$$

where

$$(\mathbf{A}_r^l)^*\mathbf{A}^l := \langle(\mathbf{A}_r)^{\otimes l}, \mathbf{A}^{\otimes l}\rangle_{3,6,\dots,3l} \in \mathbb{R}^{[nr \times nr \times n \times n]\circ l} \qquad (40)$$

Now, $\langle\mathbf{A}_r^*\mathbf{A}, \text{mat}(x_0)\,\text{mat}(x_0)^\top - M^*\rangle$ is an $nr \times nr$ matrix, so the above tensor is simply a vector outer product, being symmetric by definition. Consequently, $\mathbf{w}_1 = \mathbf{w}_0 - \eta\nabla h^l(\mathbf{w}_0)$ is still symmetric, since the addition of symmetric tensors maintains symmetric property. This completes the proof of the initial step.

Then, we proceed to show the induction step. Assume that $\mathbf{w}_{t-1}$ is symmetric, meaning that

$$\mathbf{w}_{t-1} = \sum_{j=1}^{r_m} \lambda_j (x_j^{t-1})^{\otimes l}, \quad x_j^{t-1} \in \mathbb{R}^{nr}$$

where $r_m$ is the symmetric rank of $\mathbf{w}_{t-1}$. This means that

$$\nabla h^l(\mathbf{w}_{t-1}) = \sum_{j_1, j_2, j_3}^{r_m, r_m, r_m} \lambda_{j_1}\lambda_{j_2}\lambda_{j_3}(\langle\mathbf{A}_r^*\mathbf{A}, \text{mat}(x_{j_1}^{t-1})\,\text{mat}(x_{j_2}^{t-1})^\top\rangle x_{j_3}^{t-1})^{\otimes l} -$$
$$\sum_{j_3}^{r_m} \lambda_{j_3}(\langle\mathbf{A}_r^*\mathbf{A}, M^*\rangle x_{j_3}^{t-1})^{\otimes l}$$

which again is a weighted sum of rank-1 symmetric tensors, thus being symmetric. This shows that $\mathbf{w}_t = \mathbf{w}_{t-1} - \eta\nabla h^l(\mathbf{w}_{t-1})$ is also symmetric, concluding the induction step, thereby proving the claim. $\qquad\square$

Next, we show the breakdown of tensors along the GD trajectory

**Lemma 12.** *The GD trajectory of* (5) $\{\mathbf{w}_t\}_{t=0}^{\infty}$ *admits the following breakdown for an arbitrary $t$:*

$$\mathbf{w}_{t+1} = \langle\mathbf{Z}_t, \mathbf{w}_0\rangle - \mathbf{E}_t := \tilde{\mathbf{w}}_t - \mathbf{E}_t \qquad (41)$$

*where*

$$\mathbf{Z}_t := (\mathcal{I} + \eta\langle(\mathbf{A}_r^l)^*\mathbf{A}^l, (M^*)^{\otimes l}\rangle)^t$$

$$\mathbf{E}_t := \sum_{i=1}^{t}(\mathcal{I} + \eta\langle(\mathbf{A}_r^l)^*\mathbf{A}^l, (M^*)^{\otimes l}\rangle)^{t-i}\hat{\mathbf{E}}_i$$

$$\hat{\mathbf{E}}_i := \eta\langle\langle(\mathbf{A}_r^l)^*\mathbf{A}^l, \langle\mathbf{P}(\mathbf{w}_{i-1}), \mathbf{P}(\mathbf{w}_{i-1})\rangle_{2*[l]}\rangle, \mathbf{w}_{i-1}\rangle_{2*[l]}$$

*and where $(\mathbf{A}_r^l)^*\mathbf{A}^l := \langle(\mathbf{A}_r)^{\otimes l}, \mathbf{A}^{\otimes l}\rangle_{3,6,\dots,3l} \in \mathbb{R}^{[nr \times nr \times n \times n]\circ l}$.*

*Proof of Lemma 12.* For this proof, we will proceed by induction. For $t = 1$, we have that

$$\mathbf{w}_1 = (\mathcal{I} + \eta\langle(\mathbf{A}_r^l)^*\mathbf{A}^l, (M^*)^{\otimes l} - \langle\mathbf{P}(\mathbf{w}_0), \mathbf{P}(\mathbf{w}_0)\rangle\rangle)\mathbf{w}_0$$
$$= (\mathcal{I} + \eta\langle(\mathbf{A}_r^l)^*\mathbf{A}^l, (M^*)^{\otimes l}\rangle)\mathbf{w}_0 - \eta\langle(\mathbf{A}_r^l)^*\mathbf{A}^l, \langle\mathbf{P}(\mathbf{w}_0), \mathbf{P}(\mathbf{w}_0)\rangle\rangle\mathbf{w}_0$$
$$= \langle\mathbf{Z}_1, \mathbf{w}_0\rangle - \mathbf{E}_1$$

Then, we move on to the induction step, while first assuming that it holds for some $t$. One can write

$$
\begin{aligned}
\mathbf{w}_{t+1} &= (\mathcal{I} + \eta\langle(\mathbf{A}_r^l)^*\mathbf{A}^l, (M^*)^{\otimes l} - \langle\mathbf{P}(\mathbf{w}_t), \mathbf{P}(\mathbf{w}_t)\rangle\rangle)\mathbf{w}_t \\
&= (\mathcal{I} + \eta\langle(\mathbf{A}_r^l)^*\mathbf{A}^l, (M^*)^{\otimes l}\rangle)\mathbf{w}_t - \eta\langle(\mathbf{A}_r^l)^*\mathbf{A}^l, \langle\mathbf{P}(\mathbf{w}_t), \mathbf{P}(\mathbf{w}_t)\rangle\rangle\mathbf{w}_t \\
&= (\mathcal{I} + \eta\langle(\mathbf{A}_r^l)^*\mathbf{A}^l, (M^*)^{\otimes l}\rangle)\mathbf{w}_t - \hat{\mathbf{E}}_{t+1} \\
&= (\mathcal{I} + \eta\langle(\mathbf{A}_r^l)^*\mathbf{A}^l, (M^*)^{\otimes l}\rangle)\left(\tilde{\mathbf{w}}_t - \sum_{i=1}^{t}(\mathcal{I} + \eta\langle(\mathbf{A}_r^l)^*\mathbf{A}^l, (M^*)^{\otimes l}\rangle)^{t-i}\hat{\mathbf{E}}_i\right) - \hat{\mathbf{E}}_{t+1} \\
&= \tilde{\mathbf{w}}_{t+1} - \sum_{i=1}^{t}(\mathcal{I} + \eta\langle(\mathbf{A}_r^l)^*\mathbf{A}^l, (M^*)^{\otimes l}\rangle)^{t+1-i}\hat{\mathbf{E}}_i - \hat{\mathbf{E}}_{t+1} \\
&= \tilde{\mathbf{w}}_{t+1} - \sum_{i=1}^{t+1}(\mathcal{I} + \eta\langle(\mathbf{A}_r^l)^*\mathbf{A}^l, (M^*)^{\otimes l}\rangle)^{t+1-i}\hat{\mathbf{E}}_i \\
&= \tilde{\mathbf{w}}_{t+1} - \mathbf{E}_t
\end{aligned}
$$

$\square$

Following the second step in the main outline, we aim to bound the spectral norm of $\mathbf{E}_t$, via the next lemma.

**Lemma 13.** *Given a tensor $\mathbf{E}_t$ defined in Lemma 12, assume that $\mathbf{w}_0 = \epsilon x_0^{\otimes l}$, where $\epsilon \in \mathbb{R}$ is the initialization scale. For every $t \leq t_s$,*

$$
\|\mathbf{E}_t\|_S \leq \frac{8}{r_U^l \sigma_1(U)^l}\epsilon^3(nL_s)^{l/2}(1 + \tilde{\eta}\sigma_1(U)^l)^{3t}\|x_0^{\otimes l}\|_*^3 \tag{42}
$$

*with*

$$
t_s = \left\lfloor \frac{\ln\left(\frac{\sigma_1^l(U)r_U^l}{8r^l L_s^{l/2}\|x_0^{\otimes l}\|_*^3}\frac{|x_0^\top v_1|^l}{n^{l/2}}\right) - 2\ln(\epsilon)}{2\ln(1 + \tilde{\eta}\sigma_1^l(U))} \right\rfloor \tag{43}
$$

*where $U = \langle\mathbf{A}_r^*\mathbf{A}, M^*\rangle \in \mathbb{R}^{nr \times nr}$, $r_U$ being the rank of $U$, and $\tilde{\eta} = r_U^l\eta$. $\sigma_1(U)$ denotes the largest singular value of $U$, and $v_1$ being its associated singular vector.*

*Proof of Lemma 13.* From Lemma 9 and the definition in Lemma 12, it is apparent that

$$
\|\mathbf{E}_t\|_S \leq \sum_{i=1}^{t}\|(\mathcal{I} + \eta\langle(\mathbf{A}_r^l)^*\mathbf{A}^l, (M^*)^{\otimes l}\rangle)^{t-i}\|_S\|\hat{\mathbf{E}}_i\|_* \tag{44}
$$

We proceed to derive upper bounds on the norm terms separately, and then combine them together later. We first deal with $\|\hat{\mathbf{E}}_i\|_*$. By Lemma 9, we have that

$$
\|\hat{\mathbf{E}}_i\|_* \leq \eta\|\langle(\mathbf{A}_r^l)^*\mathbf{A}^l, \langle\mathbf{P}(\mathbf{w}_{i-1}), \mathbf{P}(\mathbf{w}_{i-1})\rangle\rangle\|_*\|\mathbf{w}_{i-1}\|_*
$$

Now, assume that $\mathbf{w}_{i-1}$ admits the following breakdown

$$
\mathbf{w}_{i-1} = \sum_{j=1}^{r_{i-1}}\lambda_j(x_j^{i-1})^{\otimes l}, \quad x_j^{i-1} \in \mathbb{R}^{nr}, \ \|x_j^{i-1}\|_2 = 1 \tag{45}
$$

where $\|\mathbf{w}_{i-1}\|_* = \sum_j|\lambda_j|$. Therefore,

$$
\langle\mathbf{P}(\mathbf{w}_{i-1}), \mathbf{P}(\mathbf{w}_{i-1})\rangle = \sum_{j_1, j_2}^{r_{i-1}, r_{i-1}}\lambda_{j_1}\lambda_{j_2}\langle\mathbf{P}((x_{j_1}^{i-1})^{\otimes l}), \mathbf{P}((x_{j_2}^{i-1})^{\otimes l})\rangle,
$$

leading to

$$
\langle(\mathbf{A}_r^l)^*\mathbf{A}^l, \langle\mathbf{P}(\mathbf{w}_{i-1}), \mathbf{P}(\mathbf{w}_{i-1})\rangle\rangle = \sum_{j_1, j_2}^{r_{i-1}, r_{i-1}}\lambda_{j_1}\lambda_{j_2}\langle(\mathbf{A}_r^l)^*\mathbf{A}^l, \langle\mathbf{P}((x_{j_1}^{i-1})^{\otimes l}), \mathbf{P}((x_{j_2}^{i-1})^{\otimes l})\rangle\rangle.
$$

For given indices $j_1, j_2$ index, it follows from Lemma A.4 that

$$\langle (\mathbf{A}_r^l)^* \mathbf{A}^l, \langle \mathbf{P}((x_{j_1}^{i-1})^{\otimes l}), \mathbf{P}((x_{j_2}^{i-1})^{\otimes l}) \rangle \rangle = (\langle \mathbf{A}_r^* \mathbf{A}, \mathrm{mat}(x_{j_1}^{i-1}) \mathrm{mat}(x_{j_2}^{i-1})^\top \rangle)^{\otimes l}$$

Now, according to the definition of $\mathbf{A}_r := I_r \oslash_{1,2} \mathbf{A}$, where $\oslash$ denotes the kronecker product (a reshaped tensor vector product, where the subscript denotes the dimension with which kronecker product is applied with respect to $\mathbf{A}$), we know that

$$\langle \mathbf{A}_r^* \mathbf{A}, \mathrm{mat}(x_{j_1}^{i-1}) \mathrm{mat}(x_{j_2}^{i-1})^\top \rangle = I_r \oslash \langle \mathbf{A}^* \mathbf{A}, \mathrm{mat}(x_{j_1}^{i-1}) \mathrm{mat}(x_{j_2}^{i-1})^\top \rangle$$

Hence, the eigenvalues of the LHS are just $r$ copies of that of the RHS [44]. This further implies

$$\begin{aligned}
\|\langle \mathbf{A}_r^* \mathbf{A}, \mathrm{mat}(x_{j_1}^{i-1}) \mathrm{mat}(x_{j_2}^{i-1})^\top \rangle \|_* &= r \|\langle \mathbf{A}^* \mathbf{A}, \mathrm{mat}(x_{j_1}^{i-1}) \mathrm{mat}(x_{j_2}^{i-1})^\top \rangle \|_* \\
&\leq r\sqrt{n} \|\langle \mathbf{A}^* \mathbf{A}, \mathrm{mat}(x_{j_1}^{i-1}) \mathrm{mat}(x_{j_2}^{i-1})^\top \rangle \|_F \\
&\leq r\sqrt{nL_s} \| \mathrm{mat}(x_{j_1}^{i-1}) \mathrm{mat}(x_{j_2}^{i-1})^\top \|_F \\
&= r\sqrt{nL_s}
\end{aligned}$$

where the second last inequality follows from the RSS property, and the last equality follows from (45). Next, we apply Lemma 9 again with

$$\|\langle (\mathbf{A}_r^l)^* \mathbf{A}^l, \langle \mathbf{P}((x_{j_1}^{i-1})^{\otimes l}), \mathbf{P}((x_{j_2}^{i-1})^{\otimes l}) \rangle \|_* \leq (\|\langle \mathbf{A}_r^* \mathbf{A}, \mathrm{mat}(x_{j_1}^{i-1}) \mathrm{mat}(x_{j_2}^{i-1})^\top \rangle \|_*)^l \leq r^l (nL_s)^{l/2}$$

which leads to

$$\begin{aligned}
&\|\langle (\mathbf{A}_r^l)^* \mathbf{A}^l, \langle \mathbf{P}(\mathbf{w}_{i-1}), \mathbf{P}(\mathbf{w}_{i-1}) \rangle \rangle \|_* \\
&\leq \sum_{j_1, j_2}^{r_{i-1}, r_{i-1}} |\lambda_{j_1}||\lambda_{j_2}| \|\langle (\mathbf{A}_r^l)^* \mathbf{A}^l, \langle \mathbf{P}((x_{j_1}^{i-1})^{\otimes l}), \mathbf{P}((x_{j_2}^{i-1})^{\otimes l}) \rangle \|_* \\
&\leq r^l (nL_s)^{l/2} \sum_{j_1, j_2}^{r_{i-1}, r_{i-1}} |\lambda_{j_1}||\lambda_{j_2}| = r^l (nL_s)^{l/2} \|\mathbf{w}_{i-1}\|_*^2
\end{aligned}$$

This directly gives

$$\|\hat{\mathbf{E}}_i\|_* \leq \eta (r^2 nL_s)^{l/2} \|\mathbf{w}_{i-1}\|_*^3$$

Since our goal is to bound $\|\mathbf{E}_t\|_S$, we focus on $\|(\mathcal{I} + \eta \langle (\mathbf{A}_r^l)^* \mathbf{A}^l, (M^*)^{\otimes l} \rangle)^{t-i}\|_S$. Using binomial formula, we obtain that

$$(\mathcal{I} + \eta \langle (\mathbf{A}_r^l)^* \mathbf{A}^l, (M^*)^{\otimes l} \rangle)^{t-i} = \sum_{k=0}^{t-i} \binom{t-i}{k} \eta^k (\langle (\mathbf{A}_r^l)^* \mathbf{A}^l, (M^*)^{\otimes l} \rangle)^k$$

where $\langle (\mathbf{A}_r^l)^* \mathbf{A}^l, (M^*)^{\otimes l} \rangle \in \mathbb{R}^{nro2l}$, and $(\cdot)^k$ just denotes repeated multiplications along the even dimensions of the tensor, as explained in the disclaimer. To upper-bound the spectral norm of $(\mathcal{I} + \eta \langle (\mathbf{A}_r^l)^* \mathbf{A}^l, (M^*)^{\otimes l} \rangle)^{t-i}$, it is necessary to upper-bound the spectral norm of $(\langle (\mathbf{A}_r^l)^* \mathbf{A}^l, (M^*)^{\otimes l} \rangle)^k$. To do so, we use Lemma 10 to reformulate

$$\|\langle (\mathbf{A}_r^l)^* \mathbf{A}^l, (M^*)^{\otimes l} \rangle^k\|_S = \sup_{\|\mathbf{v}\|_* \leq 1} |\langle (\mathbf{A}_r^l)^* \mathbf{A}^l, (M^*)^{\otimes l} \rangle^k, \mathbf{v} \rangle|$$

Assume that the above supremum is achieved at $\mathbf{v}^*$, with nuclear norm decomposition of

$$\mathbf{v}^* = \sum_{j_v=1}^{r_v} \lambda_{j_v} x_{j_v,1} \otimes \cdots \otimes x_{j_v,2l}, \quad x_{j_v,p} \in \mathbb{R}^{nr}, \; \|x_{j_v,p}\|_2 = 1 \; \forall p \in [2l]$$

with $\sum_{j_v} |\lambda_{j_v}| = \|\mathbf{v}^*\|_* \leq 1$. Note that this decomposition is due to the fact that $\mathbf{v}$ is not necessarily symmetric. Again, by Lemma A.4,

$$\langle (\mathbf{A}_r^l)^* \mathbf{A}^l, (M^*)^{\otimes l} \rangle^k = \left[ (\langle \mathbf{A}_r^* \mathbf{A}, M^* \rangle)^k \right]^{\otimes l},$$

directly leading to

$$\|\langle (\mathbf{A}_r^l)^* \mathbf{A}^l, (M^*)^{\otimes l} \rangle^k\|_S = \sum_{j_v=1}^{r_v} |\lambda_{j_v} \prod_{p=0}^{l-1} x_{j_v,p*2}^\top \langle \mathbf{A}_r^* \mathbf{A}, M^* \rangle^k x_{j_v,p*2+1}|$$

Since

$$x_{j_v,p*2}^\top \langle \mathbf{A}_r^* \mathbf{A}, M^* \rangle^k x_{j_v,p*2+1} \leq \sigma_1^k(U)$$

this means that

$$\|\langle (\mathbf{A}_r^l)^* \mathbf{A}^l, (M^*)^{\otimes l} \rangle^k \|_S = (\sigma_1^k(U))^l \sum_{j_v=1}^{r_v} |\lambda_{j_v}| \leq \sigma_1^{kl}(U)$$

Going back to $(\mathcal{I} + \eta \langle (\mathbf{A}_r^l)^* \mathbf{A}^l, (M^*)^{\otimes l} \rangle)^{t-i}$,

$$\|(\mathcal{I} + \eta \langle (\mathbf{A}_r^l)^* \mathbf{A}^l, (M^*)^{\otimes l} \rangle)^{t-i}\|_S \leq \sum_{k=0}^{t-i} \binom{t-i}{k} \eta^k \|(\langle (\mathbf{A}_r^l)^* \mathbf{A}^l, (M^*)^{\otimes l} \rangle)^k\|_S$$

$$\leq \sum_{k=0}^{t-i} \binom{t-i}{k} \eta^k \sigma_1^{kl}(U) = (1 + \eta \sigma_1^l(U))^{t-i}.$$

Before further upper-bounding $\|\mathbf{E}_t\|_S$, we define $t_s$ in such a way that

$$\|\tilde{\mathbf{w}}_t - \mathbf{w}_t\|_* \leq \|\tilde{\mathbf{w}}_t\|_*, \quad \forall t \leq t_s \tag{46}$$

where $\tilde{\mathbf{w}}_t$ is defined in (41). We will later justify the existence of $t_s$ and derive a lower bound. If the above inequality holds true, we also have

$$\|\mathbf{w}_t\|_* \leq \|\tilde{\mathbf{w}}_t\|_* + \|\tilde{\mathbf{w}}_t - \mathbf{w}_t\|_* \leq 2\|\tilde{\mathbf{w}}_t\|_*.$$

Recall the binomial formula again and decompose $\tilde{\mathbf{w}}_t$ into

$$\tilde{\mathbf{w}}_t = \sum_{k=0}^t \binom{t}{k} \eta^k \langle (\mathbf{A}_r^l)^* \mathbf{A}^l, (M^*)^{\otimes l} \rangle^k \mathbf{w}_0 \tag{47}$$

Therefore, it follows from Lemma 9 that,

$$\|\tilde{\mathbf{w}}_{i-1}\|_* \leq \left( \sum_{k=0}^{i-1} \binom{i-1}{k} \eta^k \|\langle (\mathbf{A}_r^l)^* \mathbf{A}^l, (M^*)^{\otimes l} \rangle^k \|_* \right) \|\mathbf{w}_0\|_* \tag{48}$$

for all $i \leq t$. With the repeated application of Lemma 9, we have

$$\|\langle (\mathbf{A}_r^l)^* \mathbf{A}^l, (M^*)^{\otimes l} \rangle^k \|_* \leq (\|U\|_*)^{kl} \leq \left( r_U^l \sigma_1^l(U) \right)^k$$

Therefore, substituting back into (48) gives

$$\|\tilde{\mathbf{w}}_{i-1}\|_* \leq \left( \sum_{k=0}^t \binom{t}{k} \eta^k \left( r^l r_U^l \sigma_1^l(U) \right)^k \right) \|\mathbf{w}_0\|_* = (1 + \tilde{\eta} \sigma_1^l(U))^{i-1} \|\mathbf{w}_0\|_*$$

Next, plugging the above preparatory results into (44), we have that

$$\|\mathbf{E}_t\|_S \leq \sum_{i=1}^{t}(1+\eta\sigma_1^l(U))^{t-i}\eta(r^2nL_s)^{l/2}\|\mathbf{w}_{i-1}\|_*^3$$

$$\leq \sum_{i=1}^{t}(1+\eta\sigma_1^l(U))^{t-i}\eta(r^2nL_s)^{l/2}8\|\tilde{\mathbf{w}}_{i-1}\|_*^3$$

$$\leq 8\sum_{i=1}^{t}(1+\eta\sigma_1^l(U))^{t-i}\eta(r^2nL_s)^{l/2}(1+\tilde{\eta}\sigma_1^l(U))^{3i-3}\|\mathbf{w}_0\|_*^3$$

$$\leq 8\epsilon^3\eta(r^2nL_s)^{l/2}\sum_{i=1}^{t}(1+\tilde{\eta}\sigma_1^l(U))^{t-i}(1+\tilde{\eta}\sigma_1^l(U))^{3i-3}$$

$$= 8\epsilon^3\|x_0^{\otimes l}\|_*^3\eta(r^2nL_s)^{l/2}(1+\tilde{\eta}\sigma_1^l(U))^{t-1}\sum_{i=1}^{t}(1+\tilde{\eta}\sigma_1^l(U))^{2i-2}$$

$$= 8\epsilon^3\|x_0^{\otimes l}\|_*^3\eta(r^2nL_s)^{l/2}(1+\tilde{\eta}\sigma_1^l(U))^{t-1}\frac{(1+\tilde{\eta}\sigma_1^l(U))^{2t}-1}{(1+\tilde{\eta}\sigma_1^l(U))^2-1} \text{ (geometric sum)}$$

$$\leq 8\epsilon^3\|x_0^{\otimes l}\|_*^3\eta(r^2nL_s)^{l/2}(1+\tilde{\eta}\sigma_1^l(U))^{t-1}(1+\tilde{\eta}\sigma_1^l(U))^{2t}$$

$$\leq \frac{8\eta}{\tilde{\eta}\sigma_1^l(U)}\epsilon^3(r^2nL_s)^{l/2}(1+\tilde{\eta}\sigma_1^l(U))^{3t}\|x_0^{\otimes l}\|_*^3$$

$$= \frac{r^l 8}{r_U^l\sigma_1^l(U)}\epsilon^3(nL_s)^{l/2}(1+\tilde{\eta}\sigma_1^l(U))^{3t}\|x_0^{\otimes l}\|_*^3$$

proving the original claim of this lemma (42). Now, we give a lower bound on $t_s$. By recalling the breakdown (47), we have

$$\|\tilde{\mathbf{w}}_t\|_* \geq \|\tilde{\mathbf{w}}_t\|_S \geq \langle \tilde{\mathbf{w}}_t, v_1^{\otimes l}\rangle$$

$$= \epsilon\sum_{k=0}^{t}\binom{t}{k}\eta^k\left[|v_1^\top\langle\mathbf{A}_r^*\mathbf{A}, M^*\rangle^k x_0|\right]^l$$

$$= \epsilon\sum_{k=0}^{t}\binom{t}{k}\eta^k\left[|v_1^\top U^k x_0|\right]^l \tag{49}$$

$$= \epsilon\sum_{k=0}^{t}\binom{t}{k}\eta^k(|\sigma_1^k(U)v_1^\top x_0|)^l = \epsilon|v_1^\top x_0|^l(1+\eta\sigma_1^l(U))^t$$

with $v_1$ being the first singular vector of $I_r \oslash U$. Since the sensing matrices are assumed to be symmetric, $U$ is also symmetric, hence the singular vectors of $U^k$ coincide with those of $U$. By (42), we also know

$$\frac{\|\tilde{\mathbf{w}}_t - \mathbf{w}_t\|_*}{\|\tilde{\mathbf{w}}_t\|_*} \leq \frac{r^l 8}{r_U^l\sigma_1^l(U)}\epsilon^2\|x_0^{\otimes l}\|_*^3\frac{n^{l/2}}{(v_1^\top x_0)^l}L_s^{l/2}\frac{(1+\tilde{\eta}\sigma_1^l(U))^{3t}}{(1+\eta\sigma_1^l(U))^t}$$

Therefore, for (46) to hold true, we need the RHS of the above equation to be smaller than 1, meaning that

$$3t\ln(1+\tilde{\eta}\sigma_1^l(U)) \leq \ln\left(\frac{r_U^l\sigma_1^l(U)}{8r^l\epsilon^2L_s^{l/2}\|x_0^{\otimes l}\|_*^3}\frac{(v_1^\top x_0)^l}{n^{l/2}}\right) + t\ln(1+\eta\sigma_1^l(U))$$

This further implies that for (46) to hold, $t$ should satisfy

$$t < \frac{\ln\left(\frac{r_U^l\sigma_1^l(U)}{8r^l\epsilon^2L_s^{l/2}\|x_0^{\otimes l}\|_*^3}\frac{(v_1^\top x_0)^l}{n^{l/2}}\right)}{3\ln(1+\tilde{\eta}\sigma_1^l(U)) - \ln(1+\eta\sigma_1^l(U))} < \frac{\ln\left(\frac{r_U^l\sigma_1^l(U)}{8r^l\epsilon^2L_s^{l/2}\|x_0^{\otimes l}\|_*^3}\frac{(v_1^\top x_0)^l}{n^{l/2}}\right)}{2\ln(1+\tilde{\eta}\sigma_1^l(U))}$$

which after rearrangement gives (43). $\qquad\square$

Now, we present the proof of Lemma 2.

*Proof of Lemma 2.* Using the tensor Weyl's inequality (Lemma 11), we have that

$$\lambda_2^v(\mathbf{w}_t) \leq \lambda_2^v(\tilde{\mathbf{w}}_t) + \|\mathbf{E}_t\|_S \tag{50}$$
$$\lambda_1^v(\mathbf{w}_t) \geq \lambda_1^v(\tilde{\mathbf{w}}_t) - \|\mathbf{E}_t\|_S \tag{51}$$

The only remaining part of the proof is the characterization of $\lambda_1^v(\tilde{\mathbf{w}}_t)$ and $\lambda_2^v(\tilde{\mathbf{w}}_t)$. The first term is easy because we already have the characterization from the proof of Lemma 13, with (49) giving rise to

$$\|\tilde{\mathbf{w}}_t\|_S \geq \epsilon |v_1^\top x_0|^l (1 + \eta \sigma_1^l(U))^t$$

Also, by the definition of v-eigenvalues and (47), we have that

$$\lambda_2^v(\tilde{\mathbf{w}}_t) = \max_{\substack{V \\ \dim(V)=2}} \min_{\substack{v \in V \\ \|v\|_2=1}} \epsilon \sum_{k=0}^t \binom{t}{k} \eta^k \left[ |v^\top \langle \mathbf{A}_r^* \mathbf{A}, M^* \rangle^k x_0| \right]^l$$

$$= \epsilon \|x_0\|_2^l \max_{\substack{V \\ \dim(V)=2}} \min_{\substack{v \in V \\ \|v\|_2=1}} \sum_{k=0}^t \binom{t}{k} \eta^k |v^\top U^k \frac{x_0}{\|x_0\|_2}|^l$$

$$\leq \epsilon \|x_0\|_2^l \max_{\substack{V \\ \dim(V)=2}} \min_{\substack{v \in V \\ \|v\|_2=1}} \sum_{k=0}^t \binom{t}{k} \eta^k |v^\top U^k v|^l$$

$$= \epsilon \|x_0\|_2^l \sum_{k=0}^t \binom{t}{k} \eta^k |v_2^\top U^k v_2|^l$$

$$= \epsilon \|x_0\|_2^l \sum_{k=0}^t \binom{t}{k} \eta^k |\sigma_2^k(U)|^l$$

$$= \epsilon \|x_0\|_2^l (1 + \eta \sigma_2^l(U))^t$$

where $v_2$ is the singular vector associated with $\sigma_2^k(U) \; \forall k \in [t]$. Finally, combining the above equations yields (6) after rearrangements. $\qquad\square$

Next, we present a supporting lemma which explains that Gaussian concentration is suited for our purpose.

**Lemma 14.** *Let $x_0 = v_1 + g \in \mathbb{R}^{nr}$, where g is a vector with each entry being i.i.d sampled from Gaussian distribution $\mathcal{N}(0, \rho)$. For some universal constant C, the follwoing inequalities hold:*

$$\mathbb{P}\left[ |v_1^\top x_0|^l \geq (1 - \mathcal{O}(\sqrt{\rho}))^l \right] \geq 1 - 2\exp(-C/\rho),$$
$$\mathbb{P}\left[ \|x_0\|_2^l \leq (\sqrt{1 + \rho^2 nr} + \mathcal{O}(\rho^{3/2}))^l \right] \geq 1 - 2\exp(-C/\rho)$$

*Proof of Lemma 14.* We know that

$$|v_1^\top x_0| = |1 + v_1^\top g| \geq 1 - |v_1^\top x_0|$$

Theorem 2.6.3 of [52] (general Hoeffding's) gives that with probability at least $1 - 2\exp(-t^2/\rho^2)$,

$$|v^\top g| \leq t \quad \forall \|v\|_2 = 1$$

which leads to the first concentration bound after substituting $t = \mathcal{O}(\sqrt{\rho})$ with some constant $c_1$. Then, Theorem 3.1.1 in [52] gives

$$\mathbb{P}\left[ |\|x_0\|_2 - \sqrt{1 + \rho^2 nr}| \leq t \right] \geq 1 - 2\exp(-c_2 t^2/\rho^4)$$

for $g \sim \mathcal{N}(0, \rho I_{nr})$ and some constant $c_2$. This is because $\mathbb{E}[\|x_0\|_2^2] = 1 + \rho^2 nr$. Substituting $t = \mathcal{O}(\rho^{3/2})$ yields that

$$\mathbb{P}\left[ \|x_0\|_2 \leq \sqrt{1 + \rho^2 nr} + \mathcal{O}(\rho^{3/2}) \right] \geq 1 - 2\exp(-c_2/\rho)$$

which results in the second bound. Now, we choose $C = \min\{c_1, c_2\}$. $\qquad\square$

Then, we prove our main theorem of this section.

*Proof of Theorem 1.* First, set $2\zeta = \kappa$, implying that $\zeta < 1/2$. We aim to derive sufficient conditions for the following inequalities to hold:

$$\lambda_2^v(\tilde{\mathbf{w}}_t) \leq \frac{\zeta}{2}\lambda_1^v(\tilde{\mathbf{w}}_t), \tag{52}$$

$$\|\mathbf{E}_t\|_s \leq \frac{\zeta}{2}\lambda_1^v(\tilde{\mathbf{w}}_t) \tag{53}$$

By recalling Lemma 2, a sufficient condition for (52) is that

$$\epsilon\|x_0\|_2^l(1 + \eta\sigma_2^l(U))^t \leq \frac{\zeta}{2}\epsilon|v_1^\top x_0|^l(1 + \eta\sigma_1^l(U))^t$$

implying that

$$\frac{2\|x_0\|_2^l}{\zeta|v_1^\top x_0|^l} \leq \left(\frac{1 + \eta\sigma_1^l(U)}{1 + \eta\sigma_2^l(U)}\right)^t$$

which after rearrangements gives $t \geq t(\zeta, l)$, as defined in (9). Then, we obtain a sufficient condition for (53), which by Lemma 13 is

$$\frac{8r^l}{r_U^l\sigma_1(U)^l}\epsilon^3(nL_s)^{l/2}(1 + \tilde{\eta}\sigma_1(U)^l)^{3t}\|x_0^{\otimes l}\|_*^3 \leq \frac{2}{\zeta}\epsilon|v_1^\top x_0|^l(1 + \eta\sigma_1^l(U))^t \tag{54}$$

contingent on the fact that $t \leq t_s$. Therefore, before going further, we need to verify that $t(\zeta, l) \leq t_s$ for some small enough $\epsilon$. (43) implies that a sufficient condition is

$$\ln\left(\frac{2\|x_0\|_2^l}{\zeta|v_1^\top x_0|^l}\right)\ln\left(\frac{1 + \eta\sigma_1^l(U)}{1 + \eta\sigma_2^l(U)}\right)^{-1} \leq \frac{\ln\left(\frac{\sigma_1^l(U)r_U^l}{8r^lL_s^{l/2}\|x_0^{\otimes l}\|_*^3\epsilon^2}\frac{|x_0^\top v_1|^l}{n^{l/2}}\right)}{2\ln(1 + \tilde{\eta}\sigma_1^l(U))}$$

Additionally, by leveraging the identity $x/(1 + x) \leq \ln(1 + x) \leq x$, we derive the following identity

$$\frac{\ln(1 + \tilde{\eta}\sigma_1^l(U))}{\ln\left(\frac{1 + \eta\sigma_1^l(U)}{1 + \eta\sigma_2^l(U)}\right)^{-1}} \leq \frac{r_U^l(1 + \eta\sigma_1^l(U))}{1 - (\sigma_2(U)/\sigma_1(U))^l} := \Xi \tag{55}$$

Hence,

$$2\ln\left(\frac{2\|\mathbf{w}_0\|_2^l}{\zeta|v_1^\top x_0|^l}\right)\Xi \leq \ln\left(\frac{\sigma_1^l(U)r_U^l}{8r^lL_s^{l/2}\|x_0^{\otimes l}\|_*^3\epsilon^2}\frac{|x_0^\top v_1|^l}{n^{l/2}}\right)$$

and after rearrangement gives

$$\epsilon^2 \leq \frac{\sigma_1^l(U)r_U^l}{8(r^2nL_s)^{l/2}}\frac{|x_0^\top v_1|^l}{\|x_0^{\otimes l}\|_*^3}\left(\frac{2\|x_0\|_2^l}{\zeta|v_1^\top x_0|^l}\right)^{-\Xi} \tag{56}$$

Notice that all of the above terms are independent of $\epsilon$, and are positive. Therefore, a small enough $\epsilon$ exists. Also notice that a smaller step-size $\eta$ will yield a loser bound on $\epsilon$ through the dependence of $\Xi$. Now, consider (54) again. Since $T$ is finite, a sufficient condition for (54) is

$$\epsilon^2 \leq \zeta\frac{r_U^l\sigma_1(U)^l}{16(r^2nL_s)^{l/2}}\frac{|v_1^\top x_0|^l}{\|x_0^{\otimes l}\|_*^3}\left(\frac{1 + \eta\sigma_1^l(U)}{(1 + \tilde{\eta}\sigma_1(U)^l)^3}\right)^T \tag{57}$$

which can again be achieved by setting a small enough $\epsilon$, since all other terms are positive and not dependent on it. In summary, if we choose a small constant $\epsilon$ satisfying both (56) and (57), and if $t_s \geq t_T$ (which again can be achieved via a sufficiently small $\epsilon$), it is already sufficient for both (52) and (53) to hold, thereby giving:

$$\frac{\lambda_2^v(\tilde{\mathbf{w}}_t) + \|\mathbf{E}_t\|_S}{\lambda_1^v(\tilde{\mathbf{w}}_t)} \leq \zeta$$

If $\zeta < 1/2$, this further implies

$$\lambda_1^v(\tilde{\mathbf{w}}_t) > 2\lambda_2^v(\tilde{\mathbf{w}}_t) + 2\|\mathbf{E}_t\|_S \implies \|\mathbf{E}_t\|_S \leq \frac{1}{2}\lambda_1^v(\tilde{\mathbf{w}}_t) - \lambda_2^v(\tilde{\mathbf{w}}_t) \leq \frac{1}{2}\lambda_1^v(\tilde{\mathbf{w}}_t)$$

As a result,

$$\frac{\lambda_2^v(\mathbf{w}_t)}{\lambda_1^v(\mathbf{w}_t)} \leq \frac{\lambda_2^v(\tilde{\mathbf{w}}_t) + \|\mathbf{E}_t\|_S}{\lambda_1^v(\tilde{\mathbf{w}}_t) - \|\mathbf{E}_t\|_S} \leq \frac{\zeta\lambda_1^v(\tilde{\mathbf{w}}_t)}{\lambda_1^v(\tilde{\mathbf{w}}_t)/2} = 2\zeta$$

which proves (8). $\qquad\square$

Theorem 1 can also be improved via Lemma 14 as stated below.

**Corollary 1** (Corollary to Theorem 1). *Consider the optimization problem and the GD trajectory given in Theorem 1. If additionally $x_0 = v_1 + g \in \mathbb{R}^{nr}$ and $g \sim \mathcal{N}(0, \rho I_{nr})$, then*

$$\frac{\lambda_2^v(\mathbf{w}_t)}{\lambda_1^v(\mathbf{w}_t)} \leq \kappa \quad \text{for } t \asymp \ln\left(\frac{1}{\kappa}\right) \ln\left(\frac{1 + \eta\sigma_1^l(U)}{1 + \eta\sigma_2^l(U)}\right)^{-1} \tag{58}$$

*provided that*

$$\epsilon \asymp \sqrt{\kappa/2} \frac{(\sigma_1(U)r_U)^{l/2}}{4(r^2 nL_s)^{l/4}} \left(\frac{\kappa}{4}\right)^{3\Xi/2}, \quad \text{where } \Xi := \frac{r_U^l(1 + \eta\sigma_1^l(U))}{1 - (\sigma_2(U)/\sigma_1(U))^l} \tag{59}$$

*with probability at least $1 - 2\exp(-C/\rho)$ for some universal constant $C$ as $\rho \to 0$, where $\sigma_1(U)$ and $\sigma_2(U)$ are the first two singular values of $U = \langle \mathbf{A}_r, b \rangle_3$, with $v_1$ being the associated singular vector of $\sigma_1(U)$ ($\asymp$ denotes "asymptotic to", meaning that the two terms of both sides of this symbol are of the same order of magnitude).*

*Proof of Corollary 1.* The proof is similar to that of Theorem 1 (note $\zeta = \kappa/2$), and therefore we only highlight the difference. We know that (23) holds true if

$$t \geq \ln\left(\frac{2\|x_0\|_2^l}{\zeta|v_1^\top x_0|^l}\right) \ln\left(\frac{1 + \eta\sigma_1^l(U)}{1 + \eta\sigma_2^l(U)}\right)^{-1},$$

$$\epsilon^2 \leq \frac{\sigma_1^l(U)r_U^l}{8(r^2 nL_s)^{l/2}} \frac{|x_0^\top v_1|^l}{\|x_0^{\otimes l}\|_*^3} \left(\frac{2\|x_0\|_2^l}{\zeta|v_1^\top x_0|^l}\right)^{-\Xi}$$

It results from Lemma 14 that for our choice of initialization, we have that

$$\|x_0\|_2^l \asymp \|v_1^\top x_0\|^l \asymp 1$$

with probability at least $1 - 2\exp(-C/\rho)$. Thus, as long as

$$t \asymp \ln\left(\frac{2}{\zeta}\right) \ln\left(\frac{1 + \eta\sigma_1^l(U)}{1 + \eta\sigma_2^l(U)}\right)^{-1} := t_*, \tag{60}$$

$$\epsilon \asymp \frac{(\sigma_1(U)r_U)^{l/2}}{2\sqrt{2}(r^2 nL_s)^{l/4}} \left(\frac{2}{\zeta}\right)^{-\Xi/2} \tag{61}$$

(23) will hold with high probability. Next, in order for (53) to hold for $t \asymp t_*$, we know that

$$\epsilon^2 \leq \zeta \frac{r_U^l \sigma_1(U)^l}{16(r^2 nL_s)^{l/2}} \frac{|v_1^\top x_0|^l}{\|x_0^{\otimes l}\|_*^3} \left(\frac{1 + \eta\sigma_1^l(U)}{(1 + \tilde{\eta}\sigma_1(U)^l)^3}\right)^{t_*}$$

Via the same order of magnitude argument, we know that the following condition is sufficient for (53):

$$\epsilon \asymp \sqrt{\zeta} \frac{(\sigma_1(U)r_U)^{l/2}}{4(r^2 nL_s)^{l/4}} \left(\frac{1 + \eta\sigma_1^l(U)}{(1 + \tilde{\eta}\sigma_1(U)^l)^3}\right)^{t_*/2}$$

Now,

$$\left(\frac{1 + \eta\sigma_1^l(U)}{(1 + \tilde{\eta}\sigma_1(U)^l)^3}\right)^{t_*/2} \geq \left[\frac{1}{(1 + \tilde{\eta}\sigma_1(U)^l)^3}\right]^{t_*/2}$$

$$= \exp\left(-\frac{3t_*}{2}\ln(1 + \tilde{\eta}\sigma_1(U)^l)\right)$$

$$= \exp\left(-\frac{3}{2}\ln(\frac{2}{\zeta})\frac{\ln(1 + \tilde{\eta}\sigma_1(U)^l)}{\ln\left(\frac{1 + \eta\sigma_1^l(U)}{1 + \eta\sigma_2^l(U)}\right)}\right)$$

$$\geq \exp\left(-\frac{3}{2}\ln(\frac{2}{\zeta})\Xi\right) = (\frac{\zeta}{2})^{3\Xi/2}$$

where the second equality follows from the substitution of $t_*$, and the last inequality follows from (55). As a result,

$$\epsilon \asymp \sqrt{\zeta} \frac{(\sigma_1(U)r_U)^{l/2}}{4(r^2 nL_s)^{l/4}} (\frac{\zeta}{2})^{3\Xi/2} \tag{62}$$

Therefore, taking the minimum of (61) and (62), we know that

$$\epsilon \asymp \sqrt{\zeta} \frac{(\sigma_1(U)r_U)^{l/2}}{(r^2 nL_s)^{l/4}} (\frac{\zeta}{2})^{3\Xi/2} \tag{63}$$

is sufficient for (52) and (53), leading to (58) via the same steps in the proof of Theorem 1. $\qquad\square$

## D  Additional Details for Properties of Approximate Rank-1 Tensors

We start with the proof of Proposition 1.

*Proof of Proposition 1.*  Given a symmetric tensor $\mathbf{w}$, it can be decomposed as

$$\mathbf{w} = \sum_{i=1}^{r_w} \lambda_i x_i^{\otimes l}$$

where $r_w$ is $\mathbf{w}$'s symmetric rank. Now, consider the vector $w_s \in \mathbb{R}^n$ that attains the spectral norm, meaning that $\langle \mathbf{w}, w_s^{\otimes l} \rangle = \lambda_1^v(\mathbf{w})$. One can decompose each $x_i^{\otimes l}$ into a parallel component and an orthogonal component. To be more specific,

$$x_i = x_i^s + x_i^\perp \implies x_i^{\otimes l} = (x_i^s)^{\otimes l} + \sum_{j=1}^{2^l-1} \underbrace{x_i^\perp \otimes \cdots \otimes x_i^\perp}_{j} \otimes \underbrace{x_i^s \otimes \cdots \otimes x_i^s}_{l-j}$$

and it is apparent that the second term is orthogonal to $w_s^{\otimes l}$ via Lemma A.4. Therefore, we just organize all components $w_s^{\otimes l}$ together and all orthogonal components together. By definition, the parallel component has the magnitude $\lambda_1^v(\mathbf{w})$. Also, by the definition of v-eigenvalues, $\|\mathbf{w}^\dagger\|_S \leq \lambda_2^v(\mathbf{w}_t)$ since otherwise the dominant direction of $\mathbf{w}^\dagger$ will just become the second eigenvector of $\mathbf{w}$. $\qquad\square$

We now provide the proof of Proposition 2.

*Proof of Proposition 2.*  According to (17a), the gradient of (5) with respect to $\mathbf{w}$ can be expressed as
$$\nabla h^l(\mathbf{w}) = \langle \langle (\mathbf{A}_r^l)^* \mathbf{A}^l, \langle \mathbf{P}(\mathbf{w}), \mathbf{P}(\mathbf{w}) \rangle_{2*[l]} - (M^*)^{\otimes l} \rangle, \mathbf{w} \rangle_{2*[l]} \tag{64}$$
where $(\mathbf{A}_r^l)^* \mathbf{A}^l$ is defined in (40). In light of (10), one can write
$$\langle \mathbf{P}(\mathbf{w}), \mathbf{P}(\mathbf{w}) \rangle_{2*[l]} = \langle \langle \mathbf{P}^{\otimes l}, \mathbf{P}^{\otimes l} \rangle_{2*[l]}, \mathbf{w} \otimes \mathbf{w} \rangle_{3,4,7,8,\ldots,4l-1,4l}$$
$$= \underbrace{\langle \langle \mathbf{P}^{\otimes l}, \mathbf{P}^{\otimes l} \rangle, \mathbf{w}_\sigma \otimes \mathbf{w}_\sigma \rangle}_{\mathbf{a}_1} + 2 \underbrace{\langle \langle \mathbf{P}^{\otimes l}, \mathbf{P}^{\otimes l} \rangle, \mathbf{w}_\sigma \otimes \mathbf{w}^\dagger \rangle}_{\mathbf{a}_2} + \underbrace{\langle \langle \mathbf{P}^{\otimes l}, \mathbf{P}^{\otimes l} \rangle, \mathbf{w}^\dagger \otimes \mathbf{w}^\dagger \rangle}_{\mathbf{a}_3}$$
where $\mathbf{w}_\sigma = \lambda_1^v(\mathbf{w}) \hat{w}^{\otimes l}$. Note that we have dropped the subscripts from the second line and henceforth for sake of simplicity. By using this logic, (64) can be written as
$$\nabla h^l(\mathbf{w}) = \underbrace{\langle \langle (\mathbf{A}_r^l)^* \mathbf{A}^l, \langle \mathbf{a}_1 - (M^*)^{\otimes l} \rangle, \mathbf{w}_\sigma \rangle}_{\mathbf{h}_1} + \mathbf{h}_2$$
where
$$\mathbf{h}_2 = \langle \langle (\mathbf{A}_r^l)^* \mathbf{A}^l, \mathbf{a}_1 \rangle, \mathbf{w}^\dagger \rangle + \langle \langle (\mathbf{A}_r^l)^* \mathbf{A}^l, \mathbf{a}_2 \rangle, \mathbf{w}_\sigma \rangle + \langle \langle (\mathbf{A}_r^l)^* \mathbf{A}^l, \mathbf{a}_2 \rangle, \mathbf{w}^\dagger \rangle +$$
$$\langle \langle (\mathbf{A}_r^l)^* \mathbf{A}^l, \mathbf{a}_3 \rangle, \mathbf{w}_\sigma \rangle + \langle \langle (\mathbf{A}_r^l)^* \mathbf{A}^l, \mathbf{a}_3 \rangle, \mathbf{w}^\dagger \rangle - \langle \langle (\mathbf{A}_r^l)^* \mathbf{A}^l, (M^*)^{\otimes l} \rangle, \mathbf{w}^\dagger \rangle$$
The first term can be analyzed as
$$\langle \langle (\mathbf{A}_r^l)^* \mathbf{A}^l, \mathbf{a}_1 \rangle, \mathbf{w}^\dagger \rangle = \langle (\mathbf{A}_r^l)^* \mathbf{A}^l, \langle \langle \mathbf{P}^{\otimes l}, \mathbf{P}^{\otimes l} \rangle, \mathbf{w}_\sigma \otimes \mathbf{w}_\sigma \otimes \mathbf{w}^\dagger \rangle \rangle$$
and by Lemma 9, we have that
$$\| \langle \langle (\mathbf{A}_r^l)^* \mathbf{A}^l, \mathbf{a}_1 \rangle, \mathbf{w}^\dagger \rangle \|_S \leq \| \langle \langle \mathbf{P}^{\otimes l}, \mathbf{P}^{\otimes l} \rangle, \mathbf{w}_\sigma \otimes \mathbf{w}_\sigma \otimes \mathbf{w}^\dagger \rangle \|_S \| (\mathbf{A}_r^l)^* \mathbf{A}^l \|_*$$
$$= \| \langle \mathbf{P}(\mathbf{w}_\sigma), \mathbf{P}(\mathbf{w}_\sigma) \rangle \otimes \mathbf{w}^\dagger \|_S \| (\mathbf{A}_r^l)^* \mathbf{A}^l \|_* \tag{65}$$
$$\leq \lambda_1^v(\mathbf{w})^2 \| \mathbf{w}^\dagger \|_S \| (\mathbf{A}_r^l)^* \mathbf{A}^l \|_* \leq \kappa \lambda_1^v(\mathbf{w})^3 r^l \| \mathbf{A}^* \mathbf{A} \|_*^l$$
The second inequality follows form that for all $u_1 \in \mathbb{R}^n$ and $u_2 \in \mathbb{R}^{nr}$ such that $\|u_1\|_2 = 1$ and $\|u_2\|_2 = 1$:
$$\| \langle \mathbf{P}(\mathbf{w}_\sigma), \mathbf{P}(\mathbf{w}_\sigma) \rangle \otimes \mathbf{w}^\dagger \|_S = \max_{u_1, u_2} \langle \langle \mathbf{P}(\mathbf{w}_\sigma), \mathbf{P}(\mathbf{w}_\sigma) \rangle \otimes \mathbf{w}^\dagger, u_1^{\otimes 2l} \otimes u_2^{\otimes l} \rangle$$
$$\leq \lambda_1^v(\mathbf{w})^2 (u^\top \mathrm{mat}(\hat{x}) \, \mathrm{mat}(\hat{x})^\top u)^l \| \mathbf{w}^\dagger \|_S$$
$$\leq \lambda_1^v(\mathbf{w})^2 \sigma_{\max}(\mathrm{mat}(\hat{x}))^{2l} \| \mathbf{w}^\dagger \|_S$$
$$\leq \lambda_1^v(\mathbf{w})^2 \|\hat{x}\|_2^{2l} \| \mathbf{w}^\dagger \|_S$$
$$= \lambda_1^v(\mathbf{w})^2 \| \mathbf{w}^\dagger \|_S$$

Repeating this process leads to

$$\|\mathbf{h}_2\|_S \leq (3\kappa + 3\kappa^2 + \kappa^3 + \kappa\|M^*\|_F^2)\lambda_1^v(\mathbf{w})^3 r^l \|\mathbf{A}^*\mathbf{A}\|_*^l \qquad (66)$$

Similarly, $\|\mathbf{h}_1\|_S = \mathcal{O}(\lambda_1^v(\mathbf{w})^3 r^l \|\mathbf{A}^*\mathbf{A}\|_*^l)$. Now, if we assume that $\mathbf{w}$ is an FOP of (5), it means that $\nabla h^l(\mathbf{w}) = 0$, further implying $\|\nabla h^l(\mathbf{w})\|_S = 0$, and by reverse triangle inequality,

$$0 = \|\nabla h^l(\mathbf{w})\|_S \geq |\|\mathbf{h}_1\|_S - \|\mathbf{h}_2\|_S|$$

which means that $\|\mathbf{h}_1\|_S = \|\mathbf{h}_2\|_S$. Since there always exits a small enough $\kappa$ such that $\|\mathbf{h}_2\|_S = c\|\mathbf{h}_1\|_S$ with $c < 1$, and therefore the only possibility that the above inequality holds true is that $\|\mathbf{h}_1\|_S = \|\mathbf{h}_2\|_S = 0$. This implies

$$\langle\mathbf{h}_1, u^{\otimes l}\rangle = (\langle\mathbf{A}, \mathrm{mat}(w_s)\,\mathrm{mat}(w_s)^\top - M^*\rangle^\top \langle\mathbf{A}, \mathrm{mat}(w_s)\,\mathrm{mat}(u)^\top\rangle)^l = 0 \quad \forall u \in \mathbb{R}^{nr}$$

which is equivalent to the FOP condition for (2), which is (13), meaning that $\mathrm{mat}(w_s) \in \mathbb{R}^{n \times r}$ is an FOP of (2). Note that we can always scale $\mathbf{A}$ and $b$ together so that $\|\mathbf{A}^*\mathbf{A}\|_*^l$ can be normalized to 1. $\qquad\square$

Finally, we prove the main result of this paper.

*Proof of Theorem 2.* We consider the SOP condition for (5), which is (17b) for some rank-1 tensor $\Delta$. We can express it as

$$\nabla^2 h^l(\hat{\mathbf{w}})[\Delta, \Delta] = 2\underbrace{\langle\nabla f^l(\langle\mathbf{P}(\hat{\mathbf{w}}), \mathbf{P}(\hat{\mathbf{w}})\rangle_{2*[l]}), \langle\mathbf{P}(\Delta), \mathbf{P}(\Delta)\rangle_{2*[l]}\rangle}_{\mathbf{a}_1(\hat{\mathbf{w}})} +$$

$$\underbrace{\|\langle\mathbf{A}^{\otimes l}, \langle\mathbf{P}(\hat{\mathbf{w}}), \mathbf{P}(\Delta)\rangle_{2*[l]} + \langle\mathbf{P}(\Delta), \mathbf{P}(\hat{\mathbf{w}})\rangle_{2*[l]}\rangle\|_F^2}_{\mathbf{a}_2(\hat{\mathbf{w}})}$$

Let $\Delta$ be defined identically to that in the proof of Theorem 4, meaning that $\Delta = \mathrm{vec}(U)^\otimes := u^{\otimes l}$. By the same logic of (64), we have that

$$\mathbf{a}_1(\hat{\mathbf{w}}) = \langle\langle(\mathbf{A}_r^l)^*\mathbf{A}^l, \langle\mathbf{P}(\hat{\mathbf{w}}), \mathbf{P}(\hat{\mathbf{w}})\rangle - (M^*)^{\otimes l}\rangle, \Delta \otimes \Delta\rangle$$

$$= \underbrace{\langle(\mathbf{A}_r^l)^*\mathbf{A}^l, \langle\langle\mathbf{P}^{\otimes l}, \mathbf{P}^{\otimes l}\rangle, \hat{\mathbf{w}} \otimes \hat{\mathbf{w}} \otimes \Delta \otimes \Delta\rangle\rangle}_{\mathbf{b}_1} - \langle(\mathbf{A}_r^l)^*\mathbf{A}^l, (M^*)^{\otimes l} \otimes \Delta \otimes \Delta\rangle$$

Since $\hat{\mathbf{w}}$ is a $\kappa$-rank-1 tensor, by denoting $\lambda_S \hat{x}^{\otimes l} := \mathbf{w}_\sigma$, we represent

$$\mathbf{b}_1 = \langle(\mathbf{A}_r^l)^*\mathbf{A}^l, \langle\langle\mathbf{P}^{\otimes l}, \mathbf{P}^{\otimes l}\rangle, \mathbf{w}_\sigma \otimes \mathbf{w}_\sigma \otimes \Delta \otimes \Delta\rangle\rangle +$$

$$2\underbrace{\langle(\mathbf{A}_r^l)^*\mathbf{A}^l, \langle\langle\mathbf{P}^{\otimes l}, \mathbf{P}^{\otimes l}\rangle, \mathbf{w}_\sigma \otimes \hat{\mathbf{w}}^\dagger \otimes \Delta \otimes \Delta\rangle\rangle}_{\mathbf{c}_1} +$$

$$\underbrace{\langle(\mathbf{A}_r^l)^*\mathbf{A}^l, \langle\langle\mathbf{P}^{\otimes l}, \mathbf{P}^{\otimes l}\rangle, \hat{\mathbf{w}}^\dagger \otimes \hat{\mathbf{w}}^\dagger \otimes \Delta \otimes \Delta\rangle\rangle}_{\mathbf{c}_2}$$

Hence,

$$\mathbf{a}_1(\hat{\mathbf{w}}) = \mathbf{a}_1(\mathbf{w}_\sigma) + 2\mathbf{c}_1 + \mathbf{c}_2$$

Now, we turn to $\mathbf{a}_2(\hat{\mathbf{w}})$. Since the sensing matrices are assumed to be symmetric, by (29), we have

$$\mathbf{a}_2(\hat{\mathbf{w}}) = 4\langle\langle(\mathbf{A}_r^l)^*\mathbf{A}^l, \langle\mathbf{P}(\hat{\mathbf{w}}), \mathbf{P}(\Delta)\rangle, \Delta \otimes \hat{\mathbf{w}}\rangle$$

$$= 4\underbrace{\langle(\mathbf{A}_r^l)^*\mathbf{A}^l, \langle\langle\mathbf{P}^{\otimes l}, \mathbf{P}^{\otimes l}\rangle, \hat{\mathbf{w}} \otimes \Delta \otimes \hat{\mathbf{w}} \otimes \Delta\rangle\rangle}_{\mathbf{b}_2}$$

again following the procedures in (64). Given the decomposition of $\hat{\mathbf{w}}$, we decompose $\mathbf{b}_2$ similarly to $\mathbf{b}_1$:

$$\mathbf{b}_2 = \langle(\mathbf{A}_r^l)^*\mathbf{A}^l, \langle\langle\mathbf{P}^{\otimes l}, \mathbf{P}^{\otimes l}\rangle, \mathbf{w}_\sigma \otimes \Delta \otimes \mathbf{w}_\sigma \otimes \Delta\rangle\rangle +$$

$$\underbrace{\langle(\mathbf{A}_r^l)^*\mathbf{A}^l, \langle\langle\mathbf{P}^{\otimes l}, \mathbf{P}^{\otimes l}\rangle, \mathbf{w}_\sigma \otimes \Delta \otimes \hat{\mathbf{w}}^\dagger \otimes \Delta + \hat{\mathbf{w}}^\dagger \otimes \Delta \otimes \mathbf{w}_\sigma \otimes \Delta\rangle\rangle}_{\mathbf{c}_3} +$$

$$\underbrace{\langle(\mathbf{A}_r^l)^*\mathbf{A}^l, \langle\langle\mathbf{P}^{\otimes l}, \mathbf{P}^{\otimes l}\rangle, \hat{\mathbf{w}}^\dagger \otimes \Delta \otimes \hat{\mathbf{w}}^\dagger \otimes \Delta\rangle\rangle}_{\mathbf{c}_4}$$

Combining everything together, we have

$$\nabla^2 h^l(\hat{\mathbf{w}})[\Delta, \Delta] = \mathbf{a}_1(\mathbf{w}_\sigma) + 2\mathbf{c}_1 + \mathbf{c}_2 + \mathbf{a}_2(\mathbf{w}_\sigma) + 4\mathbf{c}_3 + 4\mathbf{c}_4$$
$$= \nabla^2 h^l(\mathbf{w}_\sigma)[\Delta, \Delta] + 2\mathbf{c}_1 + \mathbf{c}_2 + 4\mathbf{c}_3 + 4\mathbf{c}_4$$

In addition, following the same procedures in (65),

$$2\mathbf{c}_1 + \mathbf{c}_2 + 4\mathbf{c}_3 + 4\mathbf{c}_4 \le (10\kappa + 5\kappa^2)\lambda_S^2 r^l \|\mathbf{A}^*\mathbf{A}\|_*^l$$

Now, since $\mathbf{w}_\sigma$ is a lifted version of FOP for (2) (via Proposition 1),

$$\nabla^2 h^l(\mathbf{w}_\sigma)[\Delta, \Delta] \le -2G^l + \frac{2}{2^{l-1}}L_s^l \lambda_r(\hat{X}\hat{X}^\top)^l$$

where $\hat{X} = \mathrm{mat}(\hat{x})$ and $G := -\lambda_{\min}(\nabla f(\hat{X}\hat{X}^\top)) \ge 0$. Remember that the choice of $\Delta$ is identical. Therefore, a sufficient condition for $\nabla^2 h^l(\hat{\mathbf{w}})[\Delta, \Delta] \le 0$ is that

$$2G^l \ge \frac{2}{2^{l-1}}L_s^l \lambda_r(\hat{X}\hat{X}^\top)^l + (10\kappa + 5\kappa^2)\lambda_S^2 r^l \|\mathbf{A}^*\mathbf{A}\|_*^l$$

We can derive another sufficient condition to the above inequality, which is

$$G \ge 2^{1/l-1}L_s\lambda_r(\hat{X}\hat{X}^\top) + (5\kappa + 5\kappa^2/2)^{1/l}\lambda_S^{2/l} r \|\mathbf{A}^*\mathbf{A}\|_*$$

since $(a+b)^{1/l} \le a^{1/l} + b^{1/l}$ for $a, b \ge 0$. Following the steps of the proof of Theorem 4, we obtain that

$$\|M^* - \hat{X}\hat{X}^\top\|_F^2 > 2^{1/l}\frac{L_s}{\alpha_s}\lambda_r(\hat{X}\hat{X}^\top)\,\mathrm{tr}(M^*) + \mathcal{O}(r\kappa^{1/l})$$

is sufficient. Note that $\|\mathbf{A}^*\mathbf{A}\|_*$ can be rescaled to 1 easily. Following the same steps, we can set

$$\beta = \frac{L_s\,\mathrm{tr}(M^*)\lambda_r(\hat{X}\hat{X}^\top)}{\alpha_s\|M^* - \hat{X}\hat{X}^\top\|_F^2 - \mathcal{O}(r\kappa^{1/l})}$$

and this leads to the desirable result. $\qquad\square$

# E   Additional Experiments

In this section, we provide some additional experiments to showcase the algorithmic regularization of GD algorithm in tensor problems like (5).

This section involves the decomposition of tensors along the optimization trajectory using a known algorithm, S-HOPM, as outlined in [23]. The S-HOPM algorithms extract the dominant rank-1 component of a given tensor, so as a first step, we apply this to tensors on the trajectory, and obtain $\mathbf{w}_1$. Subsequently, this component was subtracted from the original tensor, and the extraction procedure was repeated on the resultant tensor $\mathbf{w} - \mathbf{w}_1$ to obtain a new component $\mathbf{w}_2$. This allows us to directly compute $\frac{\|\mathbf{w}_1\|_F}{\|\mathbf{w}_2\|_F}$, in the hope to approximate $\lambda_2^v(\mathbf{w}_t)/\lambda_1^v(\mathbf{w}_t)$ for some given $t$ in the trajectory. Note that this procedure mirrors the definition of the variational eigenvalue of tensors defined in Definition 4. The main source of inaccuracy is that the S-HOPM algorithm may not find the real dominant rank-1 component, as specified in the original paper. Therefore, the metric we show below only serves as an approximation of $\lambda_2^v(\mathbf{w}_t)/\lambda_1^v(\mathbf{w}_t)$.

For a practical illustration, we focused on a problem defined in Section 6.1, characterized by a parameter $n = 8$. We were particularly interested in observing the evolution of the aforementioned ratio along the optimization trajectory during the process of gradient descent optimization. The results of this analysis are tabulated below:

| iteration | 20 | 40 | 60 | 80 | 100 | 120 | 140 | 160 | 180 |
|---|---|---|---|---|---|---|---|---|---|
| $\epsilon = 10^{-5}$ | 1.16 | 0.95 | 0.82 | 0.05 | 0.03 | 0.018 | 0.026 | 0.028 | 0.013 |
| $\epsilon = 10^{-3}$ | 0.13 | 0.43 | 0.44 | 0.031 | 0.036 | 0.0008 | 0.034 | 0.028 | 0.022 |
| $\epsilon = 0.1$ | 0.14 | 0.02 | 0.05 | 0.034 | 0.031 | 0.026 | 0.022 | 0.034 | 0.037 |

This table exhibits a notable trend where the tensor gradually exhibits more of a "rank-1" nature, aligning with the assertions made in Theorem 1. Interestingly, this behavior is observed across varying initialization scales ($\epsilon$), indicating that the phenomenon is not restricted to smaller scales, thus broadening the potential applicability of our findings.

This ratio provides meaningful insights into the training dynamics, which further substantiates the claims made under Theorem 1.

# F  Custom Algorithms

---

**Algorithm 1:** CustomGD Algorithm

---

1 **Input:** learning_rate, n, r, l, prob_params, loss, g_thres, buffer, beta, gamma, eta_0
2 **Initialize variables:** A, b, escape_saddle, buffer_limit, buffer_step
3 **Function** init(*starting_point, lr*)
4     **if** *lr ≠ 0* **then**
5        | learning_rate ← lr // Update learning rate if specified
6     **end**
7     **return** $\{'curr\_iter' : 0,' t\_noise' : 0,' curr\_w' : starting\_point\}$
8 **Function** update(*gradients, opt_state*)
9     curr_iter ← opt_state['curr_iter'] + 1
10    t_noise ← opt_state['t_noise']
11    curr_w ← opt_state['curr_w']
12    **if** $\|gradients\| < g\_thres$ *and curr_iter > 100* **then**
13       **if** *escape_saddle* **then**
14          t_noise ← curr_iter
15          w_s ← find rank 1 component of curr_w using tensor PCA
16          direction ← find the escape direction of w_s // According to Theorem 2
17          this_eta ← eta_0
18          **while** *loss(curr_w + this_eta \* direction) > loss(curr_w) + beta \* this_eta \* inner_product(gradients, direction)* **do**
19             this_eta ← this_eta \* gamma // Update eta using gamma, backtracking line search
20          **end**
21          updates ← this_eta \* direction
22          escape_saddle ← False
23       **end**
24       **else**
25          buffer_step ← buffer_step + 1
26          **if** *buffer_step == buffer_limit* **then**
27             escape_saddle ← True
28             buffer_step ← 0
29          **end**
30          updates ← -learning_rate \* gradients
31       **end**
32    **end**
33    **else**
34       escape_saddle ← False
35       updates ← -learning_rate \* gradients
36    **end**
37    **return** updates, $\{'curr\_iter' : curr\_iter,' t\_noise' : t\_noise,' curr\_w' : curr\_w + updates\}$

---

**Algorithm 2:** Tensor PCA Algorithm

---

**1** **Input:** tensor, lr, epochs, gradnorm_epsilon, lambd_v, key
**2** **Function** `tensor_PCA`(*tensor, lr, epochs, gradnorm_epsilon, lambd_v, key*)
**3**     **Function** `loss`(*eigenval_eigenvec, tensor*)
**4**         lambd, v ← eigenval_eigenvec
**5**         k ← len(tensor.shape)
**6**         **for** *each element in tensor.shape* **do**
**7**             tensor ← inner(tensor, v)
**8**         **end**
**9**         first_term ← square(lambd) * power(norm(v), 2*k)
**10**         res ← first_term - 2*lambd*tensor
**11**         **return** res
**12**     s ← tensor.shape[0]
**13**     **if** *lambd_v is None* **then**
**14**         v ← random.normal(shape=(s,)) / sqrt(s)
**15**         lambd ← 0.001 * random.normal()
**16**     **end**
**17**     **else**
**18**         lambd, v ← lambd_v
**19**     **end**
**20**     loss, grads, lambd_v ← `adam_optimize`(*(loss, (lambd, v), tensor), lr, epochs, gradnorm_epsilon*)
**21**     lambd, v ← lambd_v
**22**     sign ← sign(lambd)
**23**     **return** sign * power(abs(lambd), 1 / len(tensor.shape)) * v

---