# OpenReview forum: "Algorithmic Regularization in Tensor Optimization: Towards a Lifted Approach in Matrix Sensing"
_NeurIPS.cc/2023/Conference — NeurIPS 2023 poster_

### Official Review · Reviewer_rWmQ · 2023-07-03

**Soundness:** 4 excellent
**Presentation:** 3 good
**Contribution:** 3 good
**Rating:** 5
**Confidence:** 3

**Summary:**

This paper examines the role of gradient descent in inducing implicit regularization for tensor optimization, within the lifted matrix sensing framework. The authors show that with sufficiently small initialization, gradient descent applied to the lifted problem results in rank-1 tensors and critical points with escape directions. These findings show the significance of tensor parameterization for matrix sensing, combined with first-order methods, in achieving global optimality.

**Strengths:**

- Seems well-written overall.
- Has some useful insights, like "our findings indicate that when applying GD to the tensor optimization problem (3) an implicit bias can be detected with sufficiently small initialization points."
- And also they show that, "...when initialized at a symmetric tensor, the entire GD trajectory remains symmetric, completing the requirements", which is interesting.

**Weaknesses:**

- Needs more convincing or discussion that the theory can be useful for practical applications. I find the theory interesting, but unsure of the practicality. Of course, it's nice that they give an experiment on a two-layer neural network with quadratic activations, but I'm not sure quadratic activations have been used all that much practically. Also, the last "layer" of the neural network just sums the outputs of the "hidden" layer, which is not as general. I believe this is the biggest weakness of the paper.

- Hyperparameters for the different optimizations algorithms used are not found. This makes it hard to believe that the authors made a good effort to optimize the unlifted problem via perhaps a hyperparameter search.
    - For Section 6.2, the experiment is under "low-sample" conditions, but this needs to be quantified and perhaps an experiment where there are "high-sample" (or at least non-"low-sample") conditions may be of interest just for completeness.

- A minor point: While I did look over the proofs in the appendix, the number of pages of proof in the appendix is a bit long at 21 pages. It might help the paper to condense the proofs.

**Questions:**

- Line 27-29: What are the practical applications of two-layer neural networks with quadratic activations?
- Line 137: I'm assuming FOP stands for "first-order point", and this acronym should be explicitly clarified.
- Line 232: "fist" to "first"
- Line 278: "that wether" to "whether"

**Limitations:**

- The authors do not provide an explicit discussion of the limitations.

---

> ### Author Rebuttal · Authors · 2023-08-07
>
> We would like to thank the reviewer for the detailed comments and constructive suggestions. The following are our responses to the review comments:
>
> (1) We agree with the reviewer's observation that a two-layer neural network with quadratic activation alone is not a widely used model in modern day machine learning and solely focusing on it will not greatly benefit the machine learning community. However, the intent of this paper is not to provide a "trick" that only targets specific models, and instead we intent to theoretically prove that when optimizing over tensors for low-rank problems, which is an action that is implicitly done in deep neural networks including CNN [1], implicit regularization can be expected and it will eventually lead to better optimization results in the process. We thank the reviewer for raising this important question, and will take this opportunity to elucidate the importance of this work and its applicability in more detail:
>
> - Firstly, this paper focused on the general problem of matrix sensing, which includes many direct applications like medical imaging, matrix completion/recommendation systems, motion detection, and power system state estimation [2-4]. Since many of these applications are safety-critical, it is often desirable to search for the global optimum. It has been previously shown that a tensor parametrization can be helpful in pursuit of the global solution in a special case [5], and therefore we aim to prove that no explicit constraint is needed with this parametrization due to the implicit regularization effect. In this work, we focused on two specific problems to streamline the presentation, but the theoretical analysis can be readily extended to all matrix sensing problems, thereby providing merit to a wide variety of applications.
>
> - Secondly, this problem has been extensively studied in the literature as an instance of non-convex optimization, to further promote the understanding of non-convex optimization and the process of training (Please see Section 1.1 for more references). This is why it is important to focus on this specific benchmark problem, since it will advance the current understanding of the interplay between over-parametrization and non-convexity. We find that over-parametrization is only a part of the story, since an appropriate optimization algorithm that can induce implicit regularization is also key to mitigating non-convexity. Enlarging the parametrization space is very similar to increasing the number of layers in a neural network, and therefore this work provides a different angle to the usefulness of large/deep neural networks, even if three-layer networks are already universal approximators [6], and to the problem of how to best utilize the rich parametrization via a random initialization.
>
> We will add more discussions and simulations to the paper to show the importance in applications such as state estimation.
>
> (2) For the quadratic neural network training process, the only hyper-parameter that is involved is the initial learning rate (since we use the ADAM optimizer), which is set to the default value of 0.001. We appreciate the reviewer's suggestion and tried out the success rate for the same problem described in Table 2(a) with n=8, and obtained the following values:
>
> |  n=8, unlifted  | lr = 1e-5 | lr=1e-4 | lr = 1e-3 | lr = 1e-2 | lr = 0.1 |
> | -------- | ------- | ------- | ------- | ------- | ------- |
> | m=20   | 0 | 0 | 0 | 0 | 0 |
> | m=30 | 0.1 | 0.3 |0.3 | 0.4 |0.2|
> | m=40 |0.1 | 0.4 | 0.5 | 0.6| 0.2 |
>
> The above table shows that the initial learning rate in this setting does not affect the success rate too much, and will not make the unlifted problem have comparable performance to the lifted problem.
>
> As for the "low-sample" and "high-sample" terminology, it simply denotes the value of $m$, as $m$ is the number of observations. It is well known in the literature [2] that when $m$ is very large and the sensing matrices are sampled i.i.d. from a normal distribution, the RIP constant of the problem will be close to 0, and all local minimizers will correspond to global solutions, meaning the problem can be solved globally. We thank the reviewer for pointing it out, and we will provide more explanations in the main text should the paper be accepted.
>
> (3) We appreciate the reviewer's suggestion of condensing the appendix, and will try to make it better compartmenlized, so that readers only need to refer to short sections at a time. We also thank the reviewer for pointing out a few typos, and we will fix those as well.
>
> [1] N. Razin, A. Maman, and N. Cohen, “Implicit regularization in hierarchical tensor factorization and deep convolutional neural networks,” in International Conference on Machine Learning, pp. 18422–18462, PMLR, 2022.
>
> [2] L. T. Nguyen, J. Kim, and B. Shim, “Low-rank matrix completion: A contemporary survey,” IEEE Access, vol. 7, pp. 94215–94237, 2019.
>
> [3] Y. Shechtman, Y. C. Eldar, O. Cohen, H. N. Chapman, J. Miao, and M. Segev, “Phase retrieval with application to optical imaging: A contemporary overview,” IEEE Signal Processing Magazine, vol. 32, no. 3, pp. 87–109, 2015.
>
> [4] Y. Zhang, R. Madani, and J. Lavaei, “Conic relaxations for power system state estimation with line measurements,” IEEE Transactions on Control of Network Systems, vol. 5, no. 3, pp. 1193–1205, 2017.
>
> [5] Z. Ma, I. Molybog, J. Lavaei, and S. Sojoudi, “Over-parametrization via lifting for low-rank matrix sensing: Conversion of spurious solutions to strict saddle points,” in International Conference on Machine Learning, PMLR, 2023
>
> [6] Wang, Ming-Xi, and Yang Qu. "Approximation capabilities of neural networks on unbounded domains." Neural Networks 145 (2022): 56-67.

---

> > ### Author Response · Authors · 2023-08-14
> >
> > Dear Reviewer rWmQ,
> >
> > Firstly, I would like to express my sincere gratitude for the invaluable feedback you provided on our NeurIPS submission. We have carefully reviewed your comments and believe that we can address the major concerns in our revised paper.
> >
> > Having said that, if you have had the opportunity to go through our responses and still have any additional concerns or suggestions, we would greatly appreciate hearing them. We are committed to improving the quality of our work and would welcome the chance to further discuss any points of contention or areas of improvement with you.
> >
> > Thank you once again for your time and insights. We look forward to your continued guidance.

---

> > ### Comment · Reviewer_rWmQ · 2023-08-15
> >
> > Thank you for your response.
> >
> > (1) Your first bullet-point makes sense, thank you. For the second bullet-point, I understand and empathize with the author's appeal to focus on common benchmark problems and I agree that they should be studied. But I believe one or two benchmark problems that have been exhaustively studied can have its limitations. Both because they are exhaustively studied, and because it seems too few. I believe it would help the author's argument if there were many more (common) benchmark problems, and/or a more practical benchmark to go alongside the exhaustively studied ones.
> >
> > In the rebuttal, you mentioned:
> >
> > > "We will add more discussions and simulations to the paper to show the importance in applications such as state estimation."
> >
> > This would be great and would help convince me in the potential applications of the method.
> >
> > (2) Thank you for your experiments, they have cleared up my questions about hyperparameters.
> >
> > (3) Thank you.

---

> > > ### Author Response · Authors · 2023-08-15
> > >
> > > Thank you for your feedback and for carefully reviewing our work. We genuinely appreciate the time and effort you've invested.
> > >
> > > (1) We concur with your observation regarding the potential limitations of solely focusing on the matrix sensing benchmark problem. While this paper serves as an initial exploration in this domain, our aspirations extend to broadening the analysis to encompass a diverse array of problems, leveraging the robust theoretical underpinnings established here. The primary contributions of this paper can be distilled into two pivotal insights:
> > >
> > > - First, gradient descent, when initiated with small values, exhibits a pronounced implicit bias. This suggests that models with expansive parametrization should commence with modest initializations to avoid undesirable critical points.
> > > - Second, if the optimization trajectory remains close to a constrained representation space (e.g., rank-1 tensors as delineated in our study), it inherits several advantageous properties, akin to being wholly encapsulated within that constrained space.
> > >
> > > Building upon these foundational observations, our future endeavors will focus on devising enhanced metrics and algorithms for larger machine learning models, including neural networks.
> > >
> > > (2) We have begun exploring the use of parallel computing and high-performance machines to tackle large-scale lifted problems for more significant applications. Given that this is a new framework, developing fast, tailored algorithms is a challenge. We aim to complete simulations with larger sizes within a month and incorporate them into the next version of the paper.
> > >
> > > In summary, we hope our clarifications address your concerns. We humbly request a re-evaluation of our work, keeping these clarifications in mind. Regardless of the decision, we are grateful for your review and the insights shared.

---

> > > > ### Comment · Reviewer_rWmQ · 2023-08-16
> > > >
> > > > Thank you for your response. After reading the author's responses as well as the other reviews, I will raise my rating to 5: borderline accept. I guess one more suggestion is to consider being more explicit about the limitations.

---

> > > > > ### Author Response · Authors · 2023-08-16
> > > > >
> > > > > I'd like to express my sincere gratitude for the time and effort you've dedicated to interacting with me and re-evaluating our work. Your insights and feedback are invaluable, and I truly appreciate your thorough engagement. We will make sure to explicitly state our limitations in order to avoid confusions.

---

### Official Review · Reviewer_t1t9 · 2023-07-03

**Soundness:** 3 good
**Presentation:** 3 good
**Contribution:** 3 good
**Rating:** 6
**Confidence:** 4

**Summary:**

Studies the implicit regularization of gradient descent for a certain tensor optimization problem, obtained through the “lifted” matrix sensing framework. Lifting matrix sensing problems is a technique for transforming the original (non-convex) landscape into a new (still non-convex) landscape with favorable properties. In particular, spurious local minima are converted into strict saddle points.

The current paper establishes that, under near-zero initialization, gradient descent over the lifted matrix sensing problem leads to approximate rank one tensors until a certain time step. Furthermore, it shows that one can extract a first-order point for the original matrix sensing problem from approximately rank one first-order points of the lifted problem. Along the way, derives a lifting framework over symmetric tensors for matrix sensing problems with a ground truth matrix of arbitrary rank. Experiments corroborate the theory by demonstrating that optimizing the lifted problem may lead to better reconstruction than optimizing the original problem.

**Strengths:**

1. Establishes a novel implicit regularization of gradient descent for tensor optimization problems. In particular, as far as I am aware, this work is the first to show for the specific problem of (5) that gradient descent will lead to a trajectory that is approximately rank one until a certain time step.

2. Existing analyses of implicit regularization in tensor problems have focused on gradient flow (gradient descent with an infinitesimally small step size), while the current work supports the more realistic gradient descent with a finite step size.

3. The technical tools used, e.g. the v-eigenvalue definition, may be of use for future work on the implicit regularization in tensor problems.

4. The paper reads relatively well. Ample explanations and high-level intuitions are provided regarding the theoretical analysis.


**Weaknesses:**

1. There seem to be gaps in the theory that are not adequately addressed (or mentioned explicitly). Specifically:

    - Theorem 1 applies for some k < 1. In principle, at least from the theorem statement, it seems that k can be close to one, in which case the tensor is not really approximately rank one. For Theorem 2, k needs to be upper bounded by a quantity depending on the target matrix. The matter of when we can expect this to hold is not treated.

    - Theorem 1 and 2 establish that the tensor will be approximately rank one until some time step, and that first-order points of tensors that are approximately rank one, but do not correspond to a global minimum, have an escape direction. Yet, it is not guaranteed that we will reach a critical point while the tensor is still approximately rank one.

    - In Theorem 1, it is not specified how small the initialization scale needs to be. Does the theory suggest a scale of practical size and not exponentially small in the problem parameters?

    - In Theorem 2, is the rank one escape direction in the same direction of \hat{x}? Otherwise the rank could increase upon escaping, in which case the fact that we can map rank one tensors to first-order points of the original problem may be of less use.

    Obtaining full characterizations for the dynamics of gradient descent in non-convex problems is challenging, and it is not out of the ordinary to have gaps in the theory. Yet, I firmly believe that such gaps should be explicitly discussed and perhaps reconciled via experiments (e.g. showing that indeed the tensor stays approximately rank one until convergence).

&nbsp;

2. A large part of the motivation behind looking at the lifted problem (5) and analyzing the implicit regularization of gradient descent for it, is the possibility of achieving better reconstruction in matrix sensing problems. I find the empirical evidence provided unsatisfactory in establishing the importance of the lifted formulation. The matrix sensing experiments cover only a very specific perturbed matrix completion problem and matrix sensing problem with rank one measurements. Showing that using the lifted problem (5) is viable in more generic cases or real-world datasets can greatly boost the significance and interest of the results in my opinion.

    Furthermore, for the lifting technique to be practical, $l$ cannot be too large as the size of the tensor grows exponentially with it. It is not clear whether this is limiting for typical matrix sensing problems or not from the current evaluations.

&nbsp;

3. Empirical support for Theorem 1, which establishes that the tensor will be approximately rank one until a certain time step, is lacking. The experiments only examine the resulting performance after extracting a rank one tensor from the last gradient descent iterate. However, they do not show whether the tensor is approximately rank one as the theory suggests, or it has grown in rank (i.e. the norm of the residual after reducing the rank one component is non-negligible). If it is the former, it strengthens the viability of the theory, while if the latter holds, then it is unclear whether the theory indeed explains the observed empirical phenomenon. I recommend reporting, e.g., the norm of the tensor after reducing from it the extracted rank one component or some other measure of how close to rank one the tensor is.

Additional (more minor) comments:
- In the empirical evaluation, success rate is too crude of a measure in my opinion, as it is possible that the difference in reconstruction error is not large while the difference in success rate is. I believe comparing (normalized) reconstruction errors will allow for a better comparison of the performance of the lifted and unlifted techniques.

- I found the inner product notation (without any subindices) to be confusing. One would expect the output to be a scalar.

- Escape direction from a critical point is not formally defined.

- In Theorem 1, it is not specified what t_T stands for.

- Experiments are for a custom version of gradient descent that does not quite align with the theory. I would recommend clarifying briefly in words (even if in an appendix) what are the modifications it introduces and why, rather than just referring to pseudo-code.

- In line 124 it is claimed that the objective considered in this paper is more general than those studied in related works. However, it seems to me that the setting is simply different/incomparable rather than more general. For example, the analysis of [1] considers any differentiable (locally smooth) loss.

[1] Razin, Noam, Asaf Maman, and Nadav Cohen. "Implicit regularization in hierarchical tensor factorization and deep convolutional neural networks." International Conference on Machine Learning. PMLR, 2022.

**Questions:**

1. In the numerical experiments, is there a reason for comparing success rates as opposed to normalized reconstruction errors? How was the threshold of 0.05 chosen?

2. Why not parameterize the tensor as a symmetric rank one matrix from the get go, as opposed to optimizing over the full tensor. The former is more efficient in terms of the number of parameters and guarantees that the trajectory stays rank one. Is it a matter of optimization, i.e. that a symmetric rank one parameterization may converge to spurious local minima/non-strict saddles?


**Limitations:**

As detailed in the review above, I believe there is room for improvement in terms of explicitly addressing the limitations and gaps of the current theory.

---

> ### Author Rebuttal · Authors · 2023-08-07
>
> We would like to thank the reviewer for the detailed comments and keen suggestions.
>
> (1) Due to space constraints, some explanations were placed in the appendix. We apologize for any confusion and will address each concern:
>
> - Points 1 and 3: Theorem 1 outlines a relationship between ratio $\kappa$ and iteration $t$. $\kappa$ has no upper limit and can be achieved with enough steps, given a small initialization scale. To avoid using an arbitrarily small $\epsilon$, we presented Corollary 1 in Appendix C.2, which establishes the order of magnitude for $\epsilon$ and the number of iterations needed for random initialization. For instance, to achieve the value of $\kappa$ required in Theorem 2, we need $t$ to be on the magnitude of $\ln(||M^*||^2_F) \ln \left( \frac{1+ \eta \sigma^l_1(U)}{1+ \eta \sigma^l_2(U)}\right)^{-1}$, which has a value around 100 in the specific problem presented in Section 6.1.
>
> - Point 2: Proposition 2 shows that while the tensor remains approximately rank-1, it can reach critical points of the lifted problem, with the lifted critical point comprising a dominant term and some noise.
>
> - The rank-1 escape direction differs from $\hat{x}$ but can be computed deterministically. As we only require the tensor to be approximately rank-1, we can set the learning rate to the same order of magnitude as the desired $\kappa$.
>
> (2) We used perturbed matrix completion instead of standard matrix completion because its RIP constant can be easily calculated and better aligns with our theory. In both scenarios, the lifted formulation showed clear superiority even with $l=3$, but its high computational demands limit scaling with our current computational budget. Still, we explored various problem sizes and found the success of the lifted framework to be independent of the scale itself. We have started working on how to use parallel computing and high-performance machines to solve large-scale lifted problems. Since this is a new framework, developing fast, tailored algorithms is beyond simply modifying an off-the-shelf solver. We hope to complete the simulations with larger sizes in a month and add them to the next version of the paper.
>
> (3) We do not directly showcase the rank of the tensor along the optimization trajectory because tensor rank, unlike matrix rank, is not easily computable, being NP-Hard to approximate even when $l=3$ [1]. Furthermore, for the same tensor, its rank, symmetric rank, and rank based on v-Eigenvalue can all be different [2], so there is no standard metric. This is why we instead use success rate as a surrogate measure, for which a higher success rate means higher probability of escaping from saddle points, which is a hallmark of approximate rank-1 tensors.
>
> However, we agree with the reviewer's comment that this is an important aspect of our paper. Thus, we used the algorithm (S-HOPM) in [3] to extract the dominant rank-1 component of tensors along the trajectory (which we call $w_1$), and then subtract this main component from the tensor, and repeat the extraction procedure for $w-w_1$ to get $w_2$. We then calculate the ratio of $\frac{||w_1||_2}{||w_2||_2}$. Using the problem defined in Section 6.1 with n=8, we get the following results for the ratio along the optimization trajectory for gradient descent:
>
>
> |  iteration  | 20 | 40 | 60 | 80 | 100 | 120 | 140 | 160 | 180 |
> | -------- | ------- | ------- | ------- | ------- | ------- |------- | ------- | ------- | ------- |
> | $\epsilon=10^{-5}$   | 1.16 | 0.95 | 0.82 | 0.05 | 0.03 | 0.018 | 0.026 | 0.028 | 0.013 |
> | $\epsilon=10^{-3}$ | 0.13 | 0.43 |0.44 | 0.031 |0.036| 0.0008 |0.034 | 0.028 |0.022|
> | $\epsilon=0.1$ |0.14 | 0.02 | 0.05 | 0.034| 0.031 |0.026 | 0.022 | 0.034| 0.037 |
>
> The table shows that the tensor gradually becomes more "rank-1", complying with Theorem 1. This phenomenon is not limited to small initialization scales, which is promising. However, it's essential to note that the computed ratio is not a precise reflection of $\kappa$, as rank-1 approximation may not be globally optimal for arbitrary tensors [3]. Nevertheless, this ratio provides meaningful insights into the training dynamics, further supporting Theorem 1.
>
> (4) Regarding the minor comments/questions:
>
> - We'll use different notation for tensor inner-product.
>
> - Escape direction refers to a direction where the Hessian has a negative eigenvalue.
>
> - $t_T$ denotes the maximum allowed number of iterations, with $T$ being finite.
>
> - customGD differs from vanilla GD as it finds descent directions deterministically at a critical point.
>
> - Our techniques can analyze a wider range of tensor problems than [4], but it's not a generalization as the settings differ. We'll clarify this.
>
> - The threshold of 0.05 is chosen manually in order to avoid huge reconstruction errors to corrupt our data, especially in the sense that we do not distinguish between failed cases with reconstruction errors of 0.5 and 10.
>
> - If we parametrize the problem as a symmetric rank-1 tensor from the beginning, then no escape directions can be established using the techniques in Theorem 4, so it defeats the purpose of using a lifted framework.
>
> [1] Swernofsky, Joseph. "Tensor rank is hard to approximate." Approximation, Randomization, and Combinatorial Optimization. Algorithms and Techniques (APPROX/RANDOM 2018). Schloss Dagstuhl-Leibniz-Zentrum fuer Informatik, 2018.
>
> [2] Comon, Pierre, et al. "Symmetric tensors and symmetric tensor rank." SIAM Journal on Matrix Analysis and Applications 30.3 (2008): 1254-1279
>
> [3] E. Kofidis and P. A. Regalia, “On the best rank-1 approximation of higher-order supersymmetric tensors,” SIAM Journal on Matrix Analysis and Applications, vol. 23, no. 3, pp. 863–884, 2002.
>
> [4] Razin, Noam, Asaf Maman, and Nadav Cohen. "Implicit regularization in hierarchical tensor factorization and deep convolutional neural networks." International Conference on Machine Learning. PMLR, 2022.

---

> > ### Comment · Reviewer_t1t9 · 2023-08-14
> >
> > Thank you for the detailed response, I have read it and the other reviews carefully. It has addressed my concerns, a few notes though:
> >
> > - Note that with its current phrasing, Theorem 1 supports only the existence of a single $\kappa$, whose value is not stated, and a time iteration for which the claim holds. From your response I understand that the intention is that for any $\kappa$ there exists a time iteration $t(\kappa, l)$ such that the claim holds. Thus, I strongly recommend adapting the theorem statement accordingly (the result as currently stated is substantially weaker).
> >
> > - I appreciate the additional experiments, and acknowledge the evidence for the tensor remaining approximately rank one throughout optimization.
> >
> > - As mentioned in my original review, since success rate is too crude of a measure, e.g. it is possible that the difference in reconstruction error is not large while the difference in success rate is, I still believe comparing (normalized) reconstruction errors will allow for a better comparison of the performance of the lifted and unlifted techniques.

---

> > > ### Author Response · Authors · 2023-08-15
> > >
> > > Thank you for your thoughtful feedback and for highlighting areas of improvement in our paper. We genuinely appreciate the time and effort you've invested in reviewing our work.
> > >
> > > (1) To address your concerns, we will refine Theorem 1 to clarify that $\kappa$ can be set arbitrarily, and a sufficiently large iteration $t(\kappa,l)$ will ensure the tensor along the optimization trajectory remains $\kappa$-rank-1 from $t(\kappa,l)$ onwards. The theorem will also emphasize the provision of a lower-bound for $t(\kappa,l)$ based on problem constants. Additionally, we will elucidate the simplification of this theorem through random initialization for better clarity. Your feedback was instrumental in identifying potential areas of misinterpretation.
> > >
> > > (2) We acknowledge the reviewer's concern that simply providing the success rate might be misleading due to potential minute differences among the actual values. To address this, we re-ran the experiment described in Table 2(a) with n=8 and obtained normalized reconstruction errors. Since we cannot post images or PDFs during this rebuttal phase, we present the results in tables:
> > >
> > > |  Trials  |  1 | 2 | 3 | 4 |5 | 6 | 7 | 8 | 9 | 10 |
> > > | -------- | ------- | ------- | ------- | ------- | ------- |------- | ------- | ------- | ------- | ------- |
> > > | Unlifted   | 2.82 | 2.00 | 0.01 | 2.00 | 2.00 | 2.00 | 2.00 | 2.00 | 2.00 | 2.00 |
> > > | Lifted | 8.37 e-5 | 5.81 e-4 | 2.26 e-4 | 8.19 e-5 | 8.3 e-6 | 4.00 |3.97 e-4 | 8.21 e-5 |8.3 e-6| 3.5 e-5 |
> > >
> > > The data shows a significant gap in actual numerical values, and the choice of reconstruction threshold will not significantly affect the success rate. We chose the success rate metric because, for instance, the outlier in trial 6 of the lifted experiment would skew the average reconstruction error to 0.4 with a standard deviation of 1.2. This doesn't accurately represent the phenomenon where the lifted formulation nearly solves all instances perfectly. We appreciate the reviewer's feedback and will include more distributions of reconstruction errors in the revised version of our paper, should it be accepted.
> > >
> > > In conclusion, we hope our clarifications address your concerns. We kindly request you to re-evaluate our work in light of these explanations. Regardless of the outcome, we deeply value your insights and are grateful for your thorough review.

---

> > > > ### Comment · Reviewer_t1t9 · 2023-08-15
> > > >
> > > > Thank you for the response, it has fully addressed my concerns.

---

> > > > > ### Author Response · Authors · 2023-08-16
> > > > >
> > > > > I'd like to express my sincere gratitude for the time and effort you've dedicated to interacting with me. Your insights and elaborate feedbacks are invaluable, and are very constructive to our work in general. I truly appreciate your thorough engagement.

---

### Official Review · Reviewer_Dh4F · 2023-07-04

**Soundness:** 3 good
**Presentation:** 3 good
**Contribution:** 4 excellent
**Rating:** 7
**Confidence:** 3

**Summary:**

This paper presents a new GD algorithm that is suitable for the problem of matrix sensing. Within this algorithm a 1-rank approximation of the corresponding tensor is made. A distinctive feature of this algorithm is that some points of local minima are turned into saddle points, which improves the convergence to the global optimum.

**Strengths:**

- A new modification of the gradient descent method was invented and implemented, which gives an advantage in certain problems

- This article shows the importance and application of concepts from tensor algebra to machine learning.

- All the algorithms are described in detail in Appendix

- All the proofs are given in Appendix, and all the necessary terms are neatly given in the text of the article and Appendix.





**Weaknesses:**

- The article is overloaded with technical details and terms, many of which are only defined in appendix. This makes it difficult to understand the main idea of the article. Perhaps this style would be more suitable for people specializing in tensor algebra and matrix sensing problems.

- Only few numerical experiments have demonstrated a strong advantage of the method.


- Minor:
line 156 possible typo, must be $2d$ instead of $d$

**Questions:**

Have you tried testing the algorithm on more complex real-world problems?

**Limitations:**

The theorems mention constraints on the operator $\mathcal A$, which appears in the formulation of the matrix sensing problem. Thus, the developed method is restricted to problems that can be reduced to the matrix sensing conception and those for which certain conditions are imposed. It seems difficult to say in advance (without numerical experiments) whether the algorithm will work on this particular problem.

---

> ### Author Rebuttal · Authors · 2023-08-07
>
> We would like to thank the reviewer for the detailed comments and keen suggestions. The following are our responses to the review comments:
>
> (1) We express our gratitude to the reviewer for bringing up this concern. To enhance the accessibility of our work, we intend to streamline the main text by providing concise details and devote more space to explaining the core theorems. This adjustment will facilitate comprehension for general readers less familiar with the subject matter. Furthermore, we propose to reorganize the appendix into separate sections, each self-contained, enabling readers to explore specific topics without the need to refer to other sections except when necessary for more in-depth investigation.
>
> (2) In our experiments, we focused on two problems that we believe are important in the class of matrix sensing. Due to the computationally intensive nature of the lifted framework, we conducted experiments with problem sizes that were within the limits of our available resources. Our experiments demonstrate that small initialization with a tensor parametrization can drastically increase the probability of reaching the global optimum, and that a proper choice of algorithm is very important in the over-parametrized regime. In fact, both experiments showed meaningful improvements across a number of different settings. Nevertheless, we recognize the value in applying our technique to larger-scale problems, and we have started working on how to use parallel computing and high-performance machines to solve large-scale lifted problems. The issue is that since this is a new framework, developing fast algorithms tailored to this problem is beyond simply coding the problem and passing it to an off-the-shelf solver. We hope to complete the simulations with larger sizes in a month and add them to the next version of the paper.
>
> (3) We appreciate the reviewer for pointing out our typo on line 156.
>
> (4) Regarding the limitations, this lifted technique can to be applied to general matrix sensing problems and does not need anything to be checked in advance, as long as the problem can be written in the form of equation (2). Theorem 2 is only a sufficient condition to ensure the conversion of spurious solution, and the only requirement is that the unlifted problem should have a non-zero $\alpha_s$, which can be easily satisfied with full-rank sensing matrices. Alternatively, this non-zero $\alpha_s$ can also be expected to hold with high probability if the sensing matrices are sampled from i.i.d. Gaussians [1].
>
> [1] Candes, E. J. and Plan, Y. (2011). Tight oracle inequalities for low-rank matrix recovery from a minimal number of noisy random measurements.

---

> > ### Comment · Reviewer_Dh4F · 2023-08-16
> >
> > Thank you for the detailed response. After reading your answer and discussion with other reviewers, I am going to keep the positive score 7: Accept.

---

> > > ### Author Response · Authors · 2023-08-17
> > >
> > > Thank you for dedicating your time to review our paper and reading our response. We appreciate the suggestions you brought to the table and will take these into consideration when preparing for the next version. We really appreciate your engagement.

---

### Official Review · Reviewer_RX1r · 2023-07-27

**Soundness:** 3 good
**Presentation:** 2 fair
**Contribution:** 3 good
**Rating:** 7
**Confidence:** 3

**Summary:**

Gradient descent induces implicit regularization for tensor optimization. Specifically, it has a bias towards approximately rank-1 solution in the lifted matrix sensing framework. In Theorem 1, the authors show that the ratio between the second v-eigenvalue and the first v-eigenvalue exponentially decays to 0 in terms of the number of iterations. In theorem 2, the authors show that any first-order solution of the lifted problem that is far away from the ground truth is a saddle point.


**Strengths:**

The authors carried out complicated proof and show non-trivial results. The generalization of lifted problems from r=1 to general r is very interesting. Given that this is very theoretical, the presentation is good enough to capture the main proof strategy.

**Weaknesses:**

There is still some confusion and I hope the authors can better explain them in the paper.
1) In Theorem 2, the authors show that if the approximation error is large, then GD can always find a direction to improve. However, it is not clear whether the lower bound is actually meaningful. For example, if $M^*$ is the identity matrix in the first $r$ dimension and $\hat{X}\hat{X}^T$ is the identity matrix in the next $r$ dimension. Then I would think $\hat{X}\hat{X}^T$ is not a good estimate for $M^*$ but the approximation error is 2r may not satisfy the lower bound $\frac{L_s}{\alpha_s}r $. Since the ratio $\frac{L_s}{\alpha_s}$ relates to the RIP condition and the authors have discussed how overparameterization can relax RIP constraints, I encourage authors to provide more discussions around the lower bound and possible connection to the RIP condition.

2) On line 262, the authors mentioned that "since increasing $l$ will decrease ratio $\lambda_2^v(w_t)/\lambda_1^v(w_t)$ provided that $\sigma_1(U)\geq 1$. I am not sure how strict this condition is and my guess is some simple tricks may make this condition always hold.

3) From Theorem 1, it looks like the rank 1 regularization always holds regardless of the value of $l$, and lifting may not be necessary. Theorem 2 requires a large and odd value of $l$ which is not clear to me why this condition is needed. Some explanation would be nice.

4) When $w_t$ is approximately rank-1, what does it implies about the rank of original estimate $\hat{X}$ for general r? It is not very clear to me.

**Questions:**

line 232 typo: "fist" to "first"

**Limitations:**

The main results are theoretical results.

---

> ### Author Rebuttal · Authors · 2023-08-07
>
> We would like to thank the reviewer for the detailed comments and keen suggestions. The following are our responses to the review comments:
>
> (1) The reviewer raises a nice point regarding the specific example given. It is important to emphasize that the condition in (11) represents a sufficient condition rather than a necessary one. This implies that if the distance exceeds the RHS of (11), the conversion can be ensured based on the theory. However, even when (11) does not hold, it does not preclude the possibility of conversion. To address this concern, Theorem 7 in the appendix provides a proof that, irrespective of whether (11) is met, all spurious solutions can be converted to strict saddles as long as $||M^*||_F$ remains small. We intend to clarify this distinction more explicitly in the main text should the paper be accepted.
>
> (2) The condition $\sigma_1(U) \geq 1$ is not an important one because even if $\sigma_1(U) < 1$, it still hold that $\sigma_2^l(U) \leq \sigma_1^l(U)$, and that the RHS of (7) will be larger than 1 regardless. We thank the reviewer for pointing it out, and will delete this sentence to avoid confusion.
>
> (3) Our work builds upon [1], where the authors demonstrated that a large and odd $l$ is necessary for the conversion from spurious solutions to strict saddles. However, their theoretical analysis was limited to the special case of rank-1 matrices, a property that is difficult to maintain. In our study, we show that employing the gradient descent algorithm with a small initialization implicitly preserves this property without imposing any rank constraints. This finding is established in Theorem 1. In summary, a large $l$ is essential for the optimization landscape to possess favorable properties contingent on being rank-1, and Theorem 1 establishes that a simple algorithmic choice ensures the tensor's approximate rank-1 nature, irrespective of $l$. Furthermore, Theorem 2 demonstrates that this "approximate" rank-1 property is benign when compared to being truly rank-1.
>
> (4) Then tensor $w_t$ being approximate rank-1 has no implication on the rank of the original estimate $\hat X$, and the maximum rank of $\hat X$ is assumed to be known a priori in order for (5) to be established correctly.
>
> [1] Z. Ma, I. Molybog, J. Lavaei, and S. Sojoudi, “Over-parametrization via lifting for low-rank matrix sensing: Conversion of spurious solutions to strict saddle points,” in International Conference on Machine Learning, PMLR, 2023.

---

> > ### Comment · Reviewer_RX1r · 2023-08-17
> >
> > Thanks for the response. I will remain my score.
> > Yet for point (2), if $\sigma_1(U)<1$, then increasing $l$ may decrease convergence speed and from the later theorem, large $l$ seems to be required. I think it would be good if there is any simple solution that can address the case when $\sigma_1(U)<1$.

---

> > > ### Author Response · Authors · 2023-08-18
> > >
> > > We appreciate the reviewer's diligent review and the insightful points raised. Our perspective on $l$ is that it should be chosen such that the conversion from spurious solutions to strict saddle points is ensured. Once this is established, we can leverage this information, along with other constants, to determine the required number of iterations for tensors along the optimization trajectory to approximate rank-1, as indicated by relations (7) or (9).
> > >
> > > If, as the reviewer pointed out, $\sigma_1(U)$ is notably small, we can adjust the stepsize $\eta$ to be larger initially. This adjustment allows us to achieve a satisfactory $\kappa$ without necessitating an excessive number of iterations. Furthermore, our experiments have demonstrated that usually a choice of $l=3$ is already good enough, suggesting that there's typically no need for an exceedingly large $l$, which further addresses potential concerns when $\sigma_1(U) < 1$.
> > >
> > > We recognize the importance of this observation and will ensure its inclusion in the revised version of the paper for clarity.

---

### Decision · Program_Chairs · 2023-09-21

**Decision:**

Accept (poster)

**Comment:**

This paper examines the role of gradient descent in inducing implicit regularization for tensor optimization within the context of the lifted matrix sensing framework. The paper offers interesting insights in the usefulness of tensor overparameterization for matrix problems, which translates spurious solutions to strict saddle points. After the rebuttal process, the reviewers were able to lift some initial doubts and reached a consensus for accepting this paper.